# CityLens: Evaluating Large Vision-Language Models for Urban Socioeconomic Sensing

**Tianhui Liu**[1]    **Hetian Pang**[2]    **Xin Zhang**[2]    **Tianjian Ouyang**[2]
**Zhiyuan Zhang**[3]    **Jie Feng**[2,4†]    **Yong Li**[2†]    **Pan Hui**[1†]

[1]Information Hub, The Hong Kong University of Science and Technology (Guangzhou)
[2]Department of Electronic Engineering, BNRist, Tsinghua University
[3]School of Electronic and Information Engineering, Beijing Jiaotong University
[4]Zhongguancun Academy

## Abstract

Understanding urban socioeconomic conditions through visual data is a challenging yet essential task for sustainable urban development and policy planning. In this work, we introduce *CityLens*, a comprehensive benchmark designed to evaluate the capabilities of Large Vision-Language Models (LVLMs) in predicting socioeconomic indicators from satellite and street view imagery. We construct a multi-modal dataset covering a total of 17 globally distributed cities, spanning 6 key domains: economy, education, crime, transport, health, and environment, reflecting the multifaceted nature of urban life. Based on this dataset, we define 11 prediction tasks and utilize 3 evaluation paradigms: Direct Metric Prediction, Normalized Metric Estimation, and Feature-Based Regression. We benchmark 17 state-of-the-art LVLMs across these tasks. These make CityLens the most extensive socioeconomic benchmark to date in terms of geographic coverage, indicator diversity, and model scale. Our results reveal that while LVLMs demonstrate promising perceptual and reasoning capabilities, they still exhibit limitations in predicting urban socioeconomic indicators. CityLens provides a unified framework for diagnosing these limitations and guiding future efforts in using LVLMs to understand and predict urban socioeconomic patterns. The code and data are available at `https://github.com/tsinghua-fib-lab/CityLens`.

## 1 Introduction

Understanding the socioeconomic characteristics of urban regions is fundamental to the planning, management, and sustainability of cities. Urban socioeconomic sensing, the process of quantifying indicators such as income, education, health, and transport conditions across spatial units, plays a critical role in shaping how cities function and evolve. These indicators directly influence residents' quality of life and are deeply intertwined with key aspects of urban inequality, mobility, and resource allocation. Moreover, urban socioeconomic data serves as a cornerstone for measuring progress toward several United Nations Sustainable Development Goals (UN SDGs) (Nations, 2015). Accurate and timely information on urban disparities is essential for tracking these goals and designing effective interventions. In practice, governments and urban planners rely on socioeconomic indicators to inform a wide range of decisions—from zoning regulations and infrastructure investment to public health strategies. A better understanding of spatially resolved urban indicators empowers decision-makers to allocate resources more equitably, respond to local needs, and promote inclusive urban development.

A growing body of work has explored the use of classical deep learning methods to predict urban socioeconomic indicators. Some approaches, such as Zhou et al. (2023), leverage knowledge graphs to infer socioeconomic indicators. In parallel, researchers have explored the use of urban imagery to understand cities through their visual appearance. Methods such as Li et al. (2022), Liu et al. (2023b), Lin et al. (2024), and Yong & Zhou (2024) employ contrastive learning to generate visual representations from street view or satellite images, while others apply basic computer vision models

---

[†]Corresponding author, email: fengj12ee@hotmail.com, liyong07@tsinghua.edu.cn, panhui@ust.hk.

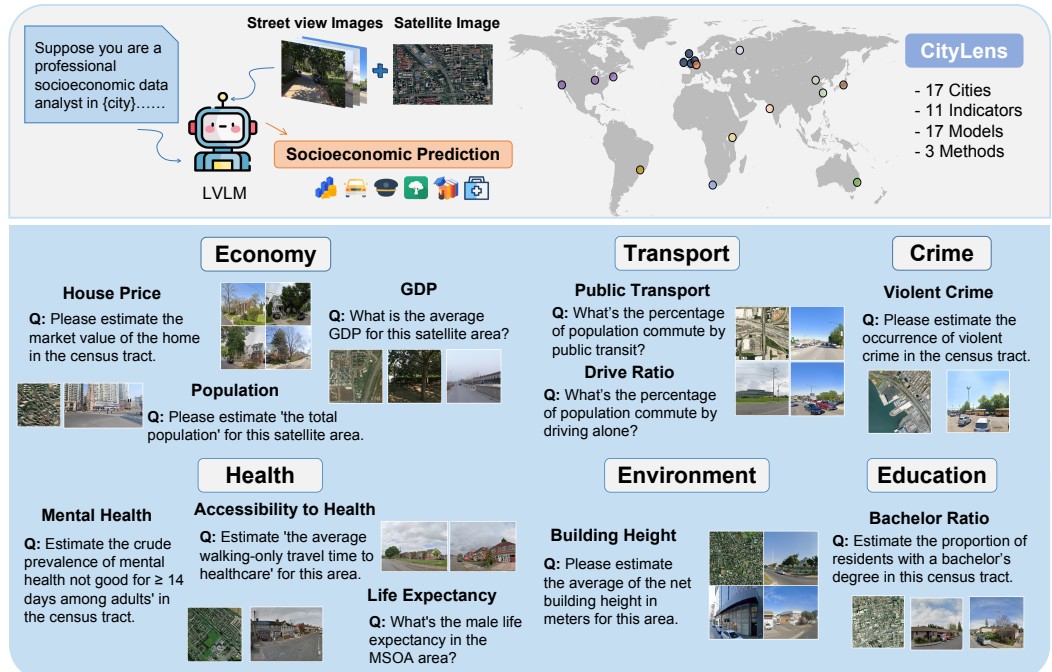

Figure 1: CityLens comprises 11 socioeconomic indicators spanning six key urban domains.

to extract visual features (Fan et al., 2023; Zhang et al., 2024). However, the classical methods face several key limitations, including difficulty in handling unstructured or multi-modal data, the inability to work across multiple countries, and cannot interpret subjective and culturally significant aspects of place. Nevertheless, large vision-language models are inherently equipped to address these challenges with their ability to integrate multiple modalities, generalize globally, and interpret cultural nuances.

In recent years, researchers have begun to leverage LVLMs and large language models (LLMs) to address some of the limitations of classical approaches (Hou et al., 2025; Yan et al., 2024; Hao et al., 2025; Manvi et al., 2024b;a). For example, Yan et al. (2024) and Hao et al. (2025) employ LVLM to generate textual descriptions from urban imagery, effectively introducing a textual modality to enrich visual understanding. Other studies, such as Manvi et al. (2024b;a), explore the ability of LLMs to predict socioeconomic indicators directly through textual prompts, and further examine issues like geographical bias across different countries. Despite these promising advances, existing works still fall short in several key aspects. Most efforts are limited in terms of spatial coverage, indicator diversity, and multi-modal integration. Crucially, there remains a lack of a systematic and unified benchmark to comprehensively evaluate how LVLMs perform across tasks, regions, and modalities in the context of urban socioeconomic sensing. To address these limitations, we propose CityLens, a comprehensive benchmark designed to evaluate the ability of large vision-language models to predict urban socioeconomic indicators using both street view and satellite imagery. CityLens spans a total of 17 cities across multiple continents, covering 11 indicators across 6 socioeconomic domains, including economy, health, education, environment, transport, and crime. By integrating diverse data modalities and global geographic coverage, CityLens enables systematic, cross-task, and cross-region evaluation of LVLMs' capabilities in urban perception, geo-visual reasoning, and numerical estimation. Overall, our contributions are summarized as follows:

- To the best of our knowledge, CityLens is the largest benchmark in urban socioeconomic sensing, along geographic coverage, indicator diversity, and model scale. It covers 17 cities across different continents, 11 indicators in 6 socioeconomic domains, using both street view and satellite imagery.
- We conduct a comprehensive evaluation of 17 state-of-the-art large vision-language models across diverse tasks and evaluation settings, systematically comparing three paradigms: Direct Metric Prediction, Normalized Estimation, and Feature-Based Regression.
- We design extensive experiments and provide detailed analysis that offers new insights into how input configuration, model architecture, and task design affect model performance, highlighting challenges, opportunities, and future directions for socioeconomic sensing with LVLMs.

## 2 METHODS

In this paper, we present CityLens, a comprehensive benchmark designed to evaluate the capabilities of large vision-language models in predicting socioeconomic indicators from both satellite and street view imagery. As illustrated in Figure 1, CityLens spans 11 real-world indicators across 6 socioeconomic domains, covering 17 globally distributed cities with diverse urban forms and development levels. To systematically assess model performance, we evaluate 17 different LVLMs using 3 distinct evaluation paradigms.

### 2.1 DATASET CONSTRUCTION

To support the evaluation of LVLMs across diverse socioeconomic indicators, as Figure 2 illustrates, we construct a region-level dataset by performing data collection, indicator selection, and data mapping. Each region is represented by 1 satellite image and 10 street view images, and is associated with corresponding socioeconomic indicator values.

**Data Collection**  We provide a detailed list of data sources for all indicators in the Appendix A.5.1; here, we briefly describe the collected indicators. Under the economy domain, we cover 7 critical indicators: Gross Domestic Product (GDP), house price, population, median household income, poverty 100%, poverty 200%, and income Gini coefficient. In the transport domain, we include seven indicators: PMT, VMT, PTRP, VTRP, walk and bike ratio, public transport ratio, and drive ratio. In the crime domain, we focus on two indicator: violent crime incidence and non-violent crime incidence, both defined as the number of crime occurrences per census tract. For the health domain, we include 9 kinds of indicators to capture different dimensions of urban health outcomes: obesity, diabetes, cancer, no leisre-time physical activity(LPA), mental health, physical health, depression rate, accessibility to healthcare, and life expectancy. In the environment domain, we consider two indicators: carbon emissions and building height. Building height is increasingly used as an explicit yet indirect indicator of urban socioeconomic development, population density, and land use intensity. Under the education domain, following Liu et al. (2023b), we use the bachelor ratio, defined as the proportion of residents holding a bachelor's degree or higher, as the target variable. These domains are selected to ensure a balanced and holistic representation of urban conditions that are commonly studied in social science and urban planning.

Since many ground-truth indicators are only available for specific countries (the US and the UK), we focus on region-level prediction tasks in three representative cities from each country. We choose New York, San Francisco, and Chicago in the US, and Leeds, Liverpool, and Birmingham in the UK. For globally available indicators, we expand coverage to cities across 6 continents, including Cape Town, Nairobi, London, Paris, Beijing, Shanghai, Moscow, Mumbai, Tokyo, Sao Paulo, and Sydney, which ensures cross-regional evaluation diversity. Beyond ground-truth indicator data, we collect both satellite images and street view images for each task region. We obtain street view images for Beijing and Shanghai using the Baidu Maps API, while for other cities, we utilize the Google Street View API. All experimental results reported in the main paper are based on these Google and Baidu sourced street view images. To promote transparency, completeness, and reproducibility of CityLens, we further construct an alternative version of the dataset using publicly accessible street view images from Mapillary, referred to as CityLens-Mapillary. We report the benchmark results based on Mapillary street view images in Appendix A.3. Additionally, the 256×256-pixel satellite images with about 4.7 m-resolution are downloaded from Esri World Imagery.

**Indicator Selection**  We initially collect ground-truth data for 28 indicators spanning 6 domains. From these, as Figure 3a illustrates, we select 11 final indicators to construct prediction tasks. The selection followed two principles: First, we assess the perceptual relevance of indicators— i.e., whether a human could reasonably infer the variable from satellite and street view imagery. Indicators such as "Estimated personal miles traveled on a working weekday", which lack visible spatial cues, are excluded. Second, we conduct Pearson correlation analysis among semantically similar indicators in the same domain to remove redundancy. For example, in the health domain, we found a high correlation between obesity and mental health (Pearson's $r = 0.7524$), which is intuitively understandable that people experiencing psychological stress or poor mental well-being tend to overeat or engage in unhealthy eating behaviors. To avoid task redundancy, we retained only mental health in the final task list.

**Data Mapping**  In CityLens, each region serves as a prediction unit, represented by 1 satellite image and 10 street view images, and is paired with a set of scalar labels corresponding to multiple target indicators. These labels are computed by mapping and aggregating raw tabular data from

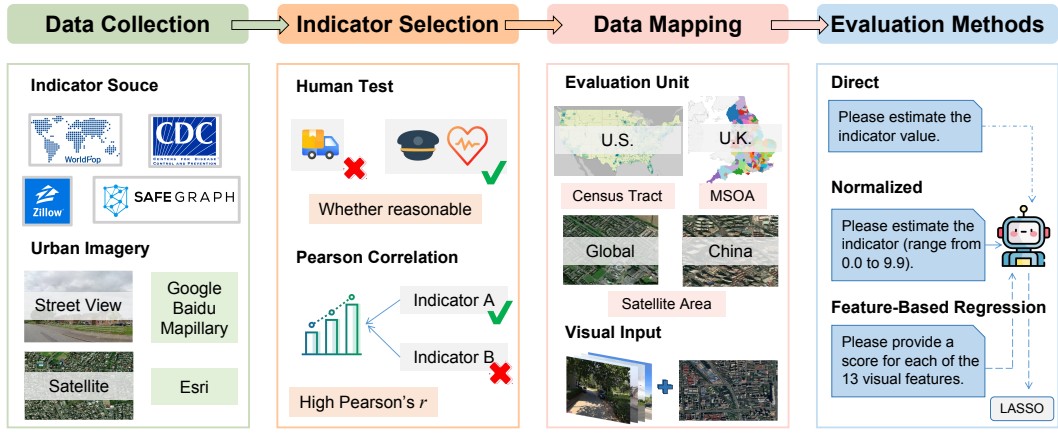

Figure 2: Benchmark Construction Pipeline, including data collection, indicator selection, data mapping and evaluation methods.

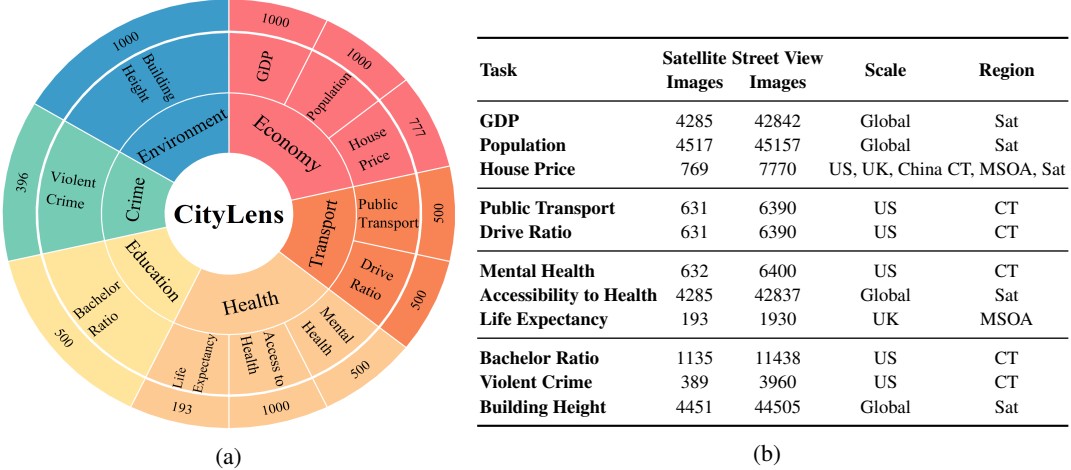

| Task | Satellite Images | Street View Images | Scale | Region |
|------|------|------|------|------|
| GDP | 4285 | 42842 | Global | Sat |
| Population | 4517 | 45157 | Global | Sat |
| House Price | 769 | 7770 | US, UK, China | CT, MSOA, Sat |
| Public Transport | 631 | 6390 | US | CT |
| Drive Ratio | 631 | 6390 | US | CT |
| Mental Health | 632 | 6400 | US | CT |
| Accessibility to Health | 4285 | 42837 | Global | Sat |
| Life Expectancy | 193 | 1930 | UK | MSOA |
| Bachelor Ratio | 1135 | 11438 | US | CT |
| Violent Crime | 389 | 3960 | US | CT |
| Building Height | 4451 | 44505 | Global | Sat |

Figure 3: (a) 11 indicators in benchmark and their counts. (b) Statistics of dataset.

heterogeneous sources to the respective region. As shown in the *region* column of Figure 3b, we define census tract-level prediction tasks for US-only indicators, and construct MSOA-level tasks for UK-only indicators using a similar strategy. For global tasks, each satellite image coverage area constitutes an evaluation unit. We first download multiple satellite images covering each city's spatial extent. Then, within each satellite image's coverage, we randomly sample 30 geographic points and collect at least 10 street view images corresponding to those points. For the China house price task, we follow the global task methodology and select Beijing and Shanghai as target cities. A detailed explanation of the mapping and aggregating of data from various sources and geographic scales for each task is provided in Appendix A.5.2. Due to resource constraints, we randomly sample up to 500 cases per task for country-specific indicators, and up to 1000 cases per task for global indicators. In practice, some tasks contain fewer samples due to data availability limitations, but these values represent the maximum sample size allowed per task. The detailed statistics of available data before applying the sampling strategy are presented in Figure 3b.

## 2.2 EVALUATION METHODOLOGIES

We design three distinct paradigms to explore the capabilities of LVLMs in socioeconomic indicator prediction. As shown in Figure 4, each paradigm is aimed at evaluating a different facet of how LVLMs can be applied to this task.

**Direct Metric Prediction**  Direct Metric Prediction refers to providing region-level urban imagery and directly querying the LVLM for the metric value, such as: "What's the percentage of the population commuting by public transit in this census tract?". In addition, the prompt positions the

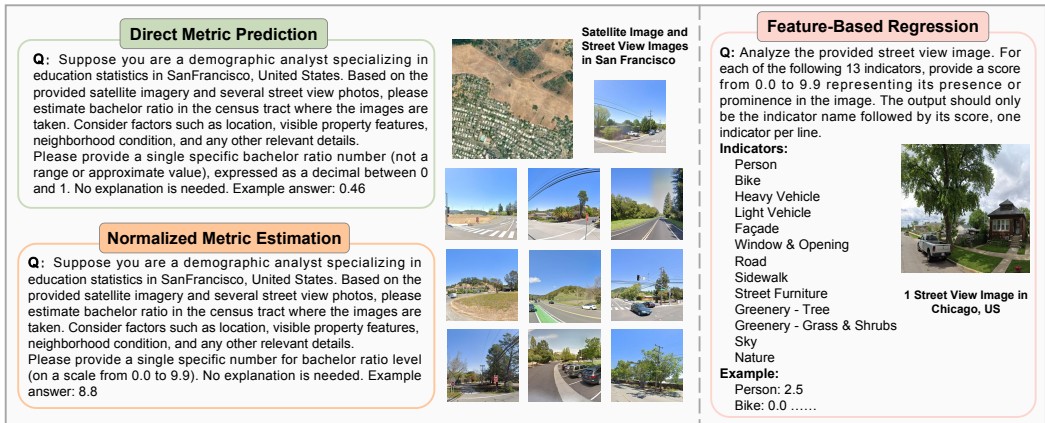

Figure 4: Prompt examples for three evaluation methodologies.

model as an urban socioeconomic scientist in a specific city. Despite this, the model faces significant challenges in accurately predicting the exact true values of these indicators.

**Normalized Metric Estimation**    Given the difficulty of directly predicting precise indicator values, we adopt a Normalized Metric Estimation approach inspired by GeoLLM (Manvi et al., 2024b). Specifically, we transform all indicator values into a normalized range from 0.0 to 9.9, discretized to one decimal place. The model is then prompted to estimate this normalized value based on the input images. This formulation aims to investigate whether the LVLM possesses coarse-grained spatial knowledge and the ability to associate visual cues with relative indicator levels.

**Feature-Based Regression**    In the Feature-Based Regression approach, we first design a structured prompt that guides the LVLM to evaluate each street view image along 13 predefined visual attributes, following the visual taxonomy proposed by Fan et al. (2023). These features capture key elements of the urban environment, such as greenery, vehicle, facade, and sidewalk. For each region, we represent its visual environment using 10 sampled street view images. For each visual feature, we compute the average score across these images, resulting in a single feature vector that characterizes the region. These aggregated visual features are then used as inputs to a LASSO regression model, which is trained to predict the ground-truth indicator values using a 5-fold cross-validation setup.

## 3    EXPERIMENTS

### 3.1    OVERALL PERFORMANCE ON FEATURE-BASED REGRESSION

**Challenge of the Benchmark for LVLMs**    As shown in Table 1, the overall performance suggests that our benchmark poses a significant challenge for current large vision-language models.  In particular, tasks such as Mental Health and Bachelor Ratio exhibit low $R^2$ scores, in some cases even approaching zero, e.g., 0.001. This highlights the difficulty of CityLens in Feature-Based Regression method:  even when leveraging visual features extracted by advanced LVLMs, the resulting representations often fail to capture the complex patterns required for accurate prediction of socioeconomic indicators. We also observe that general LVLMs underperform compared to the domain-specific contrastive learning model UrbanVLP on several tasks. Therefore, improving LVLM performance for urban socioeconomic sensing remains an important and open challenge.

**Performance Differences Across Models**    We observe substantial performance differences across LVLMs, reflecting how model scale, architecture, and training design influence their ability to extract meaningful visual features for downstream prediction. Comparing models within the same series but at different scales, we find that increasing model size does not always guarantee better performance. For example, Gemma3-12B achieves the best score on GDP and Life Expectancy, yet the 27B variant performs worse in these two tasks, with relative drops of 4.3% and 6.8% respectively. This counterintuitive result may be attributed to the unique nature of socioeconomic sensing tasks, which requires the model to consistently extract and score a predefined set of nuanced visual features from urban imagery. When comparing models from different series with similar parameter scales, clear differences emerge. For instance, Gemma3-4B significantly outperforms Qwen2.5VL-3B in nearly all

Table 1: Main results on the Feature-Based Regression method. The values in the table represent $R^2$ scores. "Mean" denotes the average performance across tasks, and "SD" refers to the standard deviation. In each row, bold indicates the best result, and underline denotes the second-best.

| Domain Tasks | GDP | Econ. Pop. | HP | Crime VC | Trans. PT | DR | Env. BH | MH | Health AH | LE | Edu. BR | Overall Mean | SD |
|---|---|---|---|---|---|---|---|---|---|---|---|---|---|
| **Baselines** | | | | | | | | | | | | | |
| **UrbanCLIP** | 0.450 | 0.030 | 0.316 | 0.033 | 0.128 | 0.123 | 0.612 | 0.021 | 0.191 | 0.024 | 0.094 | 0.184 | 0.196 |
| **UrbanVLP** | **0.717** | 0.132 | **0.559** | **0.149** | 0.551 | 0.446 | **0.807** | **0.403** | **0.382** | 0.025 | **0.422** | **0.417** | 0.243 |
| **LVLMs** | | | | | | | | | | | | | |
| **Gemma3-4B** | 0.479 | 0.252 | 0.036 | 0.103 | 0.486 | 0.365 | 0.585 | 0.183 | 0.294 | 0.148 | 0.290 | 0.293 | 0.165 |
| **Gemma3-12B** | 0.484 | 0.280 | 0.136 | 0.063 | 0.527 | 0.448 | 0.588 | 0.159 | 0.266 | **0.263** | 0.202 | 0.311 | 0.166 |
| **Gemma3-27B** | 0.463 | 0.324 | 0.141 | 0.077 | **0.567** | **0.525** | 0.590 | 0.211 | 0.283 | 0.245 | 0.297 | 0.338 | 0.166 |
| **Qwen2.5VL-3B** | 0.372 | 0.157 | 0.169 | 0.029 | 0.382 | 0.262 | 0.513 | 0.172 | 0.247 | 0.006 | 0.001 | 0.210 | 0.158 |
| **Qwen2.5VL-7B** | 0.468 | 0.304 | 0.104 | 0.053 | 0.483 | 0.308 | 0.536 | 0.166 | 0.261 | 0.119 | 0.195 | 0.272 | 0.157 |
| **Qwen2.5VL-32B** | 0.517 | 0.347 | 0.067 | 0.067 | 0.508 | 0.427 | 0.528 | 0.178 | 0.261 | 0.193 | 0.311 | 0.309 | 0.164 |
| **Llama4-Scout** | 0.460 | 0.264 | 0.164 | 0.090 | 0.508 | 0.479 | 0.524 | 0.168 | 0.280 | 0.155 | 0.197 | 0.299 | 0.155 |
| **Llama4-Maverick** | 0.452 | 0.308 | 0.233 | 0.110 | 0.547 | 0.447 | 0.523 | 0.229 | 0.293 | 0.172 | 0.249 | 0.324 | 0.139 |
| **Mistral-small-3.1-24B** | 0.452 | **0.366** | 0.144 | 0.062 | 0.499 | 0.393 | 0.571 | 0.159 | 0.260 | 0.098 | 0.198 | 0.291 | 0.166 |
| **Phi-4-multimodal** | 0.190 | 0.079 | 0.154 | 0.038 | 0.238 | 0.224 | 0.142 | 0.096 | 0.172 | 0.144 | 0.103 | 0.143 | 0.059 |
| **Nova-lite-v1** | 0.466 | 0.219 | 0.216 | 0.007 | 0.439 | 0.359 | 0.538 | 0.222 | 0.272 | 0.145 | 0.175 | 0.278 | 0.150 |
| **Minimax-01** | 0.447 | 0.336 | 0.197 | 0.068 | 0.523 | 0.448 | 0.516 | 0.113 | 0.273 | 0.162 | 0.170 | 0.295 | 0.159 |
| **Gemini-2.0-Flash** | 0.436 | 0.317 | 0.129 | 0.090 | 0.560 | 0.490 | 0.559 | 0.222 | 0.310 | 0.194 | 0.201 | 0.319 | 0.161 |
| **Gemini-2.5-Flash** | 0.375 | 0.314 | 0.143 | 0.064 | 0.527 | 0.500 | 0.568 | 0.251 | 0.277 | 0.210 | 0.203 | 0.312 | 0.156 |
| **GPT-4o-mini** | 0.425 | 0.251 | 0.119 | 0.076 | 0.470 | 0.253 | 0.554 | 0.239 | 0.295 | 0.236 | 0.163 | 0.280 | 0.141 |
| **GPT-4.1-mini** | 0.441 | 0.316 | 0.243 | 0.063 | 0.542 | 0.444 | 0.505 | 0.151 | 0.264 | 0.150 | 0.195 | 0.301 | 0.153 |
| **GPT-4.1-nano** | 0.360 | 0.314 | 0.201 | 0.084 | 0.360 | 0.198 | 0.485 | 0.175 | 0.267 | 0.086 | 0.227 | 0.251 | 0.117 |

tasks, with relative improvements ranging from 6.4% to 255% across different indicators, suggesting that Gemma's architecture or training process may enable more consistent and informative scoring of urban visual features, which in turn leads to better performance in socioeconomic prediction.

**Variations Across Different Task Types** Performance also varies across task types. Tasks like Building Height, Public Transport, and GDP tend to have relatively higher values across models, with Building Height reaching an $R^2$ of 0.590, suggesting that these indicators are associated with more observable visual cues that can be directly captured from street view images. For instance, Building Height is closely linked to the skyline and the vertical structure visible in images; Public Transport usage may be inferred from the presence of bus stops, transit signs, or road markings. In contrast, tasks such as Life Expectancy and Mental Health remain highly challenging, exhibiting low or near-zero predictive scores for many models. These indicators are influenced by latent factors such as lifestyle, stress levels, or social cohesion, which lack clear visual signals in urban imagery. Even if certain proxies exist, such as the presence of graffiti or the amount of green space, they are often subtle or semantically ambiguous, making it hard for LVLMs to interpret reliably and consistently.

## 3.2 EVALUATION OF DIRECT AND NORMALIZED ESTIMATION

**Overall Performance** We evaluate the performance of large vision-language models on all 11 tasks using both the Direct Metric Prediction and Normalized Estimation settings. To ensure meaningful analysis, we exclude model-task pairs with $R^2 \leq -0.5$ under either setting and choose to abstract away the model identity. The final comparison is visualized in Figure 5, where each point represents the performance of a specific model on a specific task, evaluated under both estimation settings. A few tasks such as House Price, Public Transport, and Building Height achieve relatively better $R^2$ scores under certain models and settings, e.g., House Price consistently exceeds 0.2 under the Direct setting. These tasks are likely more visually grounded, with cues such as building density, road layout, and commercial signage that can be directly observed from urban imagery. This suggests that some socioeconomic indicators may be approximated more easily when the visual-structural link is strong. However, the majority of results fall into the low or even negative $R^2$ range, indicating that the model's predictions often fail to explain the variance in the ground-truth indicator values. This suggests that, the models may still lack the necessary numerical grounding, contextual interpretation, and semantic alignment required to associate urban visual content with structured socioeconomic quantities. Even with normalization, which alleviates the demand for precision by coarsening the prediction space, performance remains weak across most tasks. In many cases, the model predictions tend to collapse

toward city-wide averages or exhibit a narrow output range, suggesting a lack of sensitivity to fine-grained regional variation. This behavior indicates that the models may struggle to differentiate subtle socio-spatial differences across urban regions, especially when visual cues are weak or ambiguous.

These findings highlight the inherent difficulty of CityLens; models struggle not only when asked to predict exact values, but also when the task is simplified into normalized estimation. We also incorporate the Feature-Based Regression method into a unified evaluation. A comprehensive comparison across all three paradigms is visualized in Appendix Figure 12.

**Task Preference Between Direct and Normalized Estimation**   In Figure 5, the diagonal line indicates equal performance under both methods; points above it suggest that the task benefits more from normalization, while points below indicate a preference for direct estimation. This result highlights that different tasks tend to favor different estimation strategies, depending on the nature of the indicator and its visual and semantic properties. Specifically, tasks such as Violent Crime, GDP, and Population are more frequently observed above the diagonal, suggesting that these indicators with limited direct visual correspondence benefit from a normalized formulation that emphasizes relative ranking rather than precise value prediction. These tasks are difficult to estimate accurately, but models

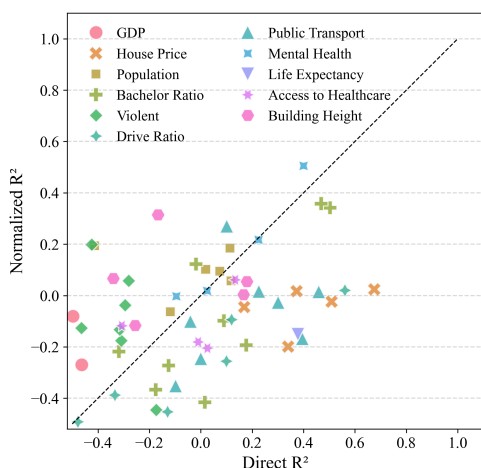

Figure 5: Comparison of task-wise $R^2$ performance between Direct Metric Prediction and Normalized Estimation across 11 socioeconomic indicators in CityLens.

may still capture coarse ordinal relationships across regions, aided by their global knowledge priors and implicit ranking sense. Conversely, tasks like Bachelor Ratio, House Price, Public Transport, and Accessibility to Health tend to fall below the diagonal, indicating better performance under the direct estimation setting. These tasks are often associated with clearer, more stable visual correlates, such as building types, infrastructure visibility, and environmental layout, which can support more precise image-to-value mappings. In addition, some indicators, e.g., Life Expectancy, exhibit narrower value ranges or lower variance, making them more amenable to direct value prediction. Moreover, for tasks like House Price and Bachelor Ratio, LVLMs may leverage latent knowledge about typical value scales across different cities, enabling surprisingly accurate numerical predictions. Taken together, these findings emphasize the importance of task-specific method selection in socioeconomic indicator prediction. The CityLens benchmark thus not only tests model capacity, but also reveals the nuanced interplay between task semantics and prediction strategy.

### 3.3 INFLUENCE OF GEOGRAPHIC VARIATION AND INPUT COMPOSITION

**City-Level Performance Variations**   To better understand the variation in socioeconomic prediction outcomes across different cities, we conduct a city-level analysis for the GDP task under the Feature-Based Regression paradigm. Each of the 13 cities is represented by 100 regions, with Gemma3-12B extracting 13 visual features per street view image. Among the 13 cities evaluated in the GDP prediction task, we observe considerable variation in model performance in Figure 6a. Cities such as Shanghai, San Francisco, and Sao Paulo achieve $R^2$ scores above 0.43, indicating relatively strong predictive performance. One possible explanation for the strong performance in cities like Shanghai lies in their well-structured urban design and high alignment between street-level appearance and economic development. These cities tend to have clear visual stratification between affluent and less affluent areas, consistent architectural patterns and homogeneous zoning that make features more learnable and high quality, diverse street view coverage. In contrast, cities like Mumbai and Moscow yield near-zero or even negative $R^2$, which may be attributed to two key factors. First, there may be a weak alignment between street-level visuals and actual economic activity, especially in cities with spatially mixed development, where wealth and poverty coexist within the same region, blurring the visual economic signal. Second, the quality and coverage of street view images can be a limiting factor. Inconsistent image sources, low resolution, or sparse sampling reduce the availability of reliable visual cues, hindering feature extraction and degrading downstream prediction.

**Impact of Input Modalities**   In this part, we evaluate the impact of input modalities by comparing model performance in three configurations: using both, only street view, and only satellite imagery.

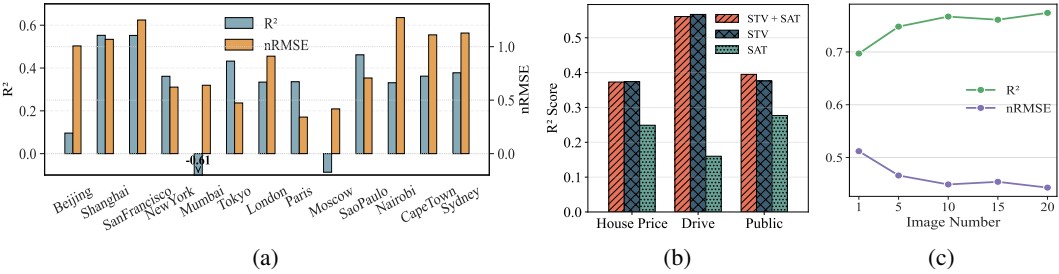

Figure 6: (a) shows the results of the GDP prediction task across 13 different cities. (b) presents the results showing that satellite imagery has limited impact on prediction. (c) demonstrates that increasing the number of street view images leads to progressive improvement in predictive performance.

We test House price, Public transport, and Drive ratio using Gemini-2.0-Flash under the Direct Metric Prediction setting. Contrary to prior findings that satellite imagery is often more discriminative than street view imagery for urban representation (Sun et al., 2025; Hao et al., 2025), our results in Figure 6b show that using street view images alone achieves performance comparable to using both street view and satellite imagery, and significantly outperforms using satellite imagery alone. This suggests that street view images provide more semantically rich and fine-grained visual cues, such as building façades, commercial signage, and infrastructure quality. These ground-level features are likely more tightly coupled with socioeconomic indicators and more readily interpreted by current LVLMs, which have been pretrained extensively on image–language pairs featuring such localized, human-centric content. While satellite imagery exhibits weaker predictive performance, it still contributes independent spatial context, such as urban morphology and building layout. However, it may not offer the same level of semantic density at the resolution used in CityLens.

**Effect of Street View Image Quantity**   To evaluate the impact of street view image quantity on prediction performance, we conduct experiments using Llama4-Maverick on the House Price task under the Direct Metric Prediction setting. Each region includes one satellite image and a varying number of street view images: 1, 5, 10, 15, or 20. We also test a no-image baseline following the design in Manvi et al. (2024b), where only the geographic coordinates and address are provided. In this setting, the model often refuses to respond, occasionally suggesting external resources like local housing websites, demonstrating both its conversational safety and limitation in open-world knowledge retrieval. From Figure 6c, we observe a clear trend: increasing the number of street view images consistently improves model performance. This suggests that a richer visual context helps the model form a more accurate understanding of the region's socioeconomic condition.

## 3.4    REASONING CAPABILITIES AND MODEL ARCHITECTURE COMPARISON

| Tasks | HP | PT | DR | BR |
|---|---|---|---|---|
| Gemma3-12B | -0.145 | 0.226 | 0.120 | -0.019 |
| **Gemma3-12B-CoT** | 0.121 | 0.156 | 0.076 | -0.049 |
| Llama4-Maverick | 0.795 | 0.459 | 0.406 | 0.503 |
| **Llama4-Maverick-CoT** | 0.794 | 0.392 | 0.270 | 0.167 |
| Gemini-2.0-Flash | 0.373 | 0.395 | 0.561 | 0.177 |
| **Gemini-2.0-Flash-CoT** | 0.602 | 0.436 | 0.508 | 0.222 |

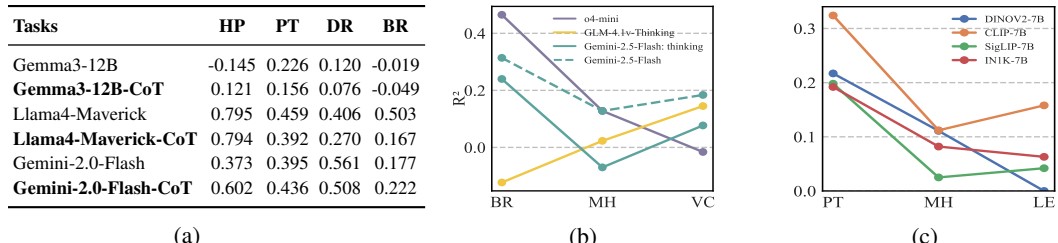

Figure 7: (a) Model performance with and without CoT prompting. (b) Performance of reasoning models. (c) Comparison of different vision encoders used within the evaluated models.

**Comparison of Chain-of-Thought Prompting vs. Standard Prompting**   Following the designs of Zhang et al. (2025b) and Xu et al. (2024), we implement a Chain-of-Thought (CoT) prompting strategy tailored to the urban socioeconomic sensing context. The example CoT prompt is shown in A.7.4. We conduct an evaluation of CoT prompting on four representative tasks using three different models under the Direct Metric Prediction setting. As the Figure 7a below shows, we observe that the effect of CoT prompting varies by task. For the House Price task, CoT almost consistently improves performance across all models, suggesting it helps with the complex reasoning involved in interpreting housing-related visual and semantic cues. In contrast, for the Drive Ratio task, CoT often reduces performance, possibly because this task relies more on direct visual features rather than step-by-step reasoning. From a model perspective, Gemini-2.0-Flash benefits most consistently

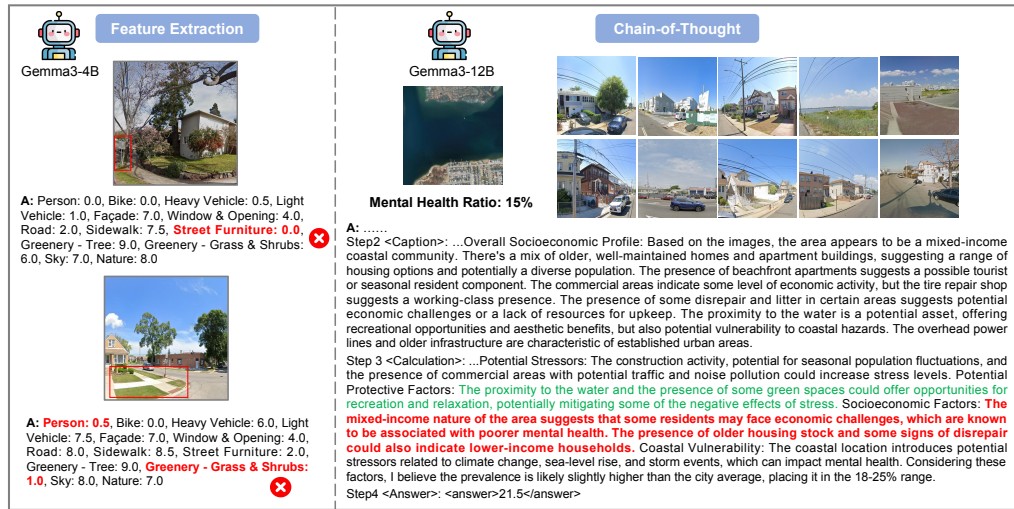

Figure 8: Representative error cases from Feature-Based Regression and CoT Prompting.

from CoT, with improvements across nearly all tasks. However, Llama4-Maverick shows performance drops with CoT. One possible explanation is that Llama4-Maverick already possesses strong internal reasoning abilities, and the externally imposed CoT structure may not align with its learned inference patterns, leading to performance degradation.

**Reasoning Model Performance with Standard Prompt**   We test three advanced reasoning models on three tasks that are inherently difficult to predict directly from images under the Normalized setting. We note that CoT prompting and reasoning models represent two distinct approaches: CoT focuses on prompt design, injecting explicit reasoning steps into the input to guide model thinking, while reasoning models are evaluated using standard prompts to assess their intrinsic reasoning capabilities. As shown in Figure 7b, none of the powerful reasoning models achieve strong performance across all tasks, highlighting the challenge of predicting abstract social indicators from visual data alone. For instance, GLM-4.1v-Thinking performs best on Violent Crime ($R^2$=0.145) but poorly on Bachelor Ratio, while o4-mini achieves the highest result on Bachelor Ratio ($R^2$=0.465) but returns a negative $R^2$ on Violent Crime. Interestingly, we observe that Gemini-2.5-Flash performs better in its non-thinking version compared to its thinking version. This may be because urban socioeconomic sensing is not a purely logical reasoning task (as in math or code), but rather requires a nuanced combination of visual understanding and contextual inference. In such settings, reasoning-specific adaptations may not always align well with the nature of the task.

**The Impact of Different Vision Encoders**   We also follow the setup in Karamcheti et al. (2024) and conduct experiments using four models, which differ only in their vision encoders, under the Feature-Based Regression setting to investigate the role of visual backbones. This result in Figure 7c suggests that LVLMs initialized with CLIP as the vision encoder produce most informative and semantically aligned outputs for urban socioeconomic sensing. The improved downstream performance may stem from CLIP's ability to extract visual cues that the language model can more effectively reason over. DINOv2 performs moderately but inconsistently, struggling on tasks like Life Expectancy, suggesting its self-supervised features may lack the semantic depth needed for abstract urban indicators. In contrast, SigLIP and IN1K perform consistently poorly, indicating that general-purpose contrastive or classification-based encoders are less effective at capturing relevant visual cues.

### 3.5  ERROR ANALYSIS AND UPPER BOUNDS OF LVLMS

**Error Cases on Challenging Tasks**   To better understand why LVLMs underperform on challenging tasks, we analyze representative errors observed under both the Feature-Based Regression and CoT Prompting setups. As illustrated in Figure 8, errors can arise from both visual perception and linguistic reasoning. For example, during feature extraction, Gemma3-4B fails to detect small but meaningful elements such as street signs, hallucinates non-existent persons, and underestimates visible greenery, assigning a low score to "Grass & Shrubs". These errors reveal a lack of fine-grained visual grounding and semantic alignment, which can propagate into downstream reasoning and prediction. We also observe reasoning errors in the CoT setting. For instance, in a Mental Health

Table 2: Results of Fine-Tuned LVLMs on the CityLens Benchmark.

| Tasks | GDP | Pop. | PT | DR | BH | MH | AH | BR |
|---|---|---|---|---|---|---|---|---|
| **Fine-tuned Qwen2.5-VL-7B** | 0.628 | 0.231 | 0.502 | 0.628 | 0.872 | 0.418 | 0.364 | 0.442 |
| **Fine-tuned Qwen3-VL-8B** | 0.626 | 0.107 | 0.545 | 0.638 | 0.869 | 0.397 | 0.304 | 0.536 |
| **Fine-tuned Llama3.2-VL-11B** | 0.562 | 0.287 | 0.348 | 0.256 | 0.829 | 0.248 | 0.256 | 0.157 |

prediction case, Gemma3-12B overly focuses on a few old and modest houses while overlooking numerous well-maintained, even upscale beachfront apartments visible in the street view images. Moreover, it fails to leverage the region's proximity to water, which is a known factor with strong aesthetic and calming effects that are directly linked to mental well-being. This suggests that current LVLMs may struggle to appropriately weigh holistic environmental cues during reasoning.

**The Potential of Fine-Tuned LVLMs**    We conduct preliminary supervised fine-tuning on Qwen2.5-VL-7B, Qwen3-VL-8B, and Llama3.2-VL-11B, using additional CityLens data not included in the benchmark under the Direct Metric Prediction setup. Since the training data lacks samples for House Price, Violent Crime, and Life Expectancy, these tasks are excluded from evaluation. While general state-of-the-art LVLMs often perform poorly on CityLens benchmark, frequently yielding near-zero or even negative $R^2$ scores, our results in Table 2 show that fine-tuned LVLMs, regardless of base model or parameter size, achieve consistently strong performance across nearly all tasks. These findings highlight the promising potential of LVLMs for urban socioeconomic sensing, and further provide a preliminary estimate of the upper bound that such models can achieve when properly adapted for this domain. This reinforces the central motivation behind CityLens and underscores the value of developing domain-specific LVLMs for addressing this socially important challenge.

## 4    RELATED WORK

**Urban Socioeconomic Sensing**    A growing number of studies have attempted to predict socioeconomic indicators in urban environments. Zhou et al. (2023) and Liu et al. (2023b) employ knowledge graph based approaches to support socioeconomic inference. Li et al. (2022) propose a contrastive learning framework based on structural urban imagery to support socioeconomic prediction. Fan et al. (2023) extract features from street view imagery via a computer vision model to predict urban indicators. More recent studies have begun incorporating LLMs into this domain (Liu et al., 2025; Ouyang et al., 2024). Yan et al. (2024) and Hao et al. (2025) combine contrastive learning on urban images with LLM-generated textual prompts. Manvi et al. (2024b) extracts geospatial knowledge from LLMs through fine-tuning and prompt design, while Li et al. (2024b) evaluates LLMs on socioeconomic tasks across region-level and city-level granularity. Different from these works, CityLens is the first benchmark to systematically evaluate the ability of LVLMs to predict socioeconomic indicators using both street view and satellite imagery.

**Benchmarking LLM and LVLM**    In recent years, LLMs have rapidly advanced in commonsense and reasoning, leading to the creation of diverse benchmarks across domains. These include benchmarks for code (Jimenez et al., 2023; Jain et al., 2024), mathematics (Zhong et al., 2023; Wang et al., 2023), city (Feng et al., 2025a), as well as agent-based tasks (Liu et al., 2023a; Qin et al., 2023). Furthermore, numerous multimodal benchmarks have also emerged for LVLMs, including comprehensive benchmarks like Li et al. (2023) and Yue et al. (2024), and domain-specific ones such as Xia et al. (2024), Zhou et al. (2025), and Hu et al. (2024). While there are some benchmarks include urban imagery (e.g., Feng et al. (2025c;b); Zhang et al. (2025a); Zhou et al. (2025)), they are not specifically designed for urban socioeconomic sensing and address only a very limited subset of such tasks. Therefore, after a comprehensive review of previous works, we propose a new benchmark to fill this gap and provide opportunities to bridge LVLMs with urban sensing applications.

## 5    CONCLUSION

In this paper, we introduce *CityLens*, a benchmark for evaluating the ability of large vision-language models to predict socioeconomic indicators from satellite and street view imagery. Through extensive experiments across 3 evaluation paradigms and 17 state-of-the-art models, we find that while current models exhibit promising perceptual abilities on certain visually grounded tasks, they still face major challenges in making accurate and generalizable predictions across domains and regions. CityLens provides a foundation for analyzing these limitations and motivates further research into enhancing the capabilities of large vision-language models in urban socioeconomic sensing.

ACKNOWLEDGEMENTS

This work was supported in part by the National Key Research and Development Program of China under Grant 2024YFC3307602, in part by the Guangdong Provincial Talent Program under Grant No.2023JC10X009, in part by the National Key Research and Development Program of China under grant 2024YFC3307603. This work was also sponsored by Tsinghua-Toyota Joint Research Institute Inter-disciplinary Program.

ETHICS STATEMENT

**Privacy**   All street view images in CityLens are sourced from platforms such as Google Street View, Baidu Maps, and Mapillary, which enforce automatic blurring of sensitive visual information, including faces and license plates. No personal or identifiable data is collected, stored, or annotated by the authors. All images are used exclusively for academic research purposes. Furthermore, the image resolution is coarse-grained, and all imagery is de-identified, ensuring that no individual-level visual content is exposed. The socioeconomic indicators used in the benchmark are aggregated at the regional level (e.g., Census Tract) rather than the individual level, further mitigating privacy risks.

**Geographic Bias and Fairness**   CityLens covers 17 cities across all six continents, ensuring a high level of geographic diversity. However, for certain indicators, particularly those culturally sensitive data, ground-truth labels are unavailable in some cities, most notably in regions within the Global South. As a result, the distribution of prediction tasks is uneven across regions, which may raise concerns of underrepresentation of Global South cities. Beyond task availability, such geographic imbalances in data coverage may also contribute to potential geographic biases in how current LVLMs generalize across diverse socioeconomic and cultural contexts. We include a preliminary bias audit in Appendix A.9, where we observe noticeable differences in performance across cities. We highlight this as both a diagnostic insight and an opportunity for future research, particularly in addressing fairness and robustness in cross-regional prediction scenarios.

**Misuse Declaimer**   CityLens is designed solely for research and evaluation purposes. It should not be used to inform real-world decisions in areas such as policing, health, or public resource allocation. The benchmark includes sensitive socioeconomic and crime-related indicators that are highly context-dependent. Any use of model outputs evaluated on CityLens for operational or policy purposes must be preceded by ethical review, fairness assessment, and domain-specific validation. We strongly discourage the use of this benchmark for surveillance or enforcement without appropriate safeguards.

REPRODUCIBILITY STATEMENT

Our work aims to ensure transparency and reproducibility through full release of both data and code. All resources are made publicly available via `https://github.com/tsinghua-fib-lab/CityLens` and `https://huggingface.co/datasets/Tianhui-Liu/CityLens-Data`.

**Dataset**   The CityLens Benchmark includes two versions of the dataset:

- CityLens (Google/Baidu-based): This version uses street view images retrieved via Google and Baidu APIs. Due to licensing restrictions, we cannot directly distribute the images. Instead, we follow the practice of Fan et al. (2023); Huang et al. (2024); Wang et al. (2025), and provide a complete list of pano ids along with download scripts for automated image retrieval. Additionally, we release the task data covering 11 socioeconomic indicators in the repository.
- CityLens-Mapillary (Open-source): This version leverages street view images from Mapillary, an open-source street-level imagery platform. All images in this version are fully accessible.

Both versions of the dataset share the same set of satellite images.

**Code**   We release all code required for data processing and evaluation. Detailed instructions and usage examples are provided in the README.md file within the repository.

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

# A APPENDIX

## A.1 THE USE OF LARGE LANGUAGE MODELS

This paper made limited use of large language models to assist with improving the clarity and stylistic quality of writing. The use of LLMs was restricted to language editing and polishing of author-written content. No text was generated solely by the model without human verification, and all technical claims, experimental results, and interpretations were entirely conceived, written, and validated by the authors. The authors retained full responsibility for the content of the paper and ensured that the use of LLMs complied with relevant ethical standards and publication guidelines.

## A.2 DISCUSSION

Our benchmark results highlight both the potential and limitations of using LVLMs to predict region-level socioeconomic indicators. While some tasks, particularly those with visually salient correlates such as building height achieve moderate performance, most indicators remain challenging to estimate accurately. Their outputs often converge around city-level averages, suggesting that the model lacks sensitivity to intra-city variation, and consequently exhibits limited geospatial grounding. In the two LVLM as predictor paradigms, we observe that different tasks tend to favor different strategies: some perform better under direct value prediction, while others benefit from normalized estimation. Overall, our results indicate that the Feature-Based Regression paradigm, where the LLM functions as a feature enhancer, significantly outperforms the two predictor-based methods. These findings suggest several promising directions for future research. First, while the feature-based method relies on a trained LASSO regressor, the predictor-based methods are initially evaluated in a zero-shot setting. Through fine-tuning LVLMs, we have demonstrated the significant potential of training domain-specific LVLMs for urban socioeconomic sensing. Second, future improvements may come from designing prompts that more closely reflect human reasoning patterns, beyond standard CoT prompting. While our current experiments already incorporate CoT prompts, we believe that further performance gains may be achieved through cognitively informed prompt design. To support this direction, we also outline a hypothesis in Appendix A.10 on how LVLMs accomplish urban socioeconomic sensing. Finally, we envision the development of a domain-specific agent framework tailored to urban socioeconomic sensing, which could combine visual perception, geospatial knowledge, and reasoning modules to make robust and context-aware predictions in real-world scenarios.

## A.3 BENCHMARK RESULTS WITH MAPILLARY STREET VIEW IMAGES

To promote transparency, completeness, and reproducibility, we construct an alternative version of CityLens using publicly available street view images from the open-source Mapillary platform, referred to as CityLens-Mapillary. This version serves as an open-source complement to the main benchmark and is intended for use in scenarios where proprietary street view APIs (Google and Baidu) are inaccessible, representing a worst-case scenario due to access restrictions or licensing limitations.

Due to the relative sparsity and inconsistency of open-source street view coverage, the number of supported tasks in CityLens-Mapillary is reduced. Table 3 summarizes the number of available prediction tasks for each indicator, comparing the original CityLens with the Mapillary-based version. Table 4 presents the results on CityLens-Mapillary. We observe a modest drop in performance compared to the original CityLens, which we attribute to the generally lower image quality and coverage of the Mapillary platform. For instance, in the UK region, predictions for Life Expectancy show several negative $R^2$ values, likely resulting from degraded visual input. While we applied light manual filtering to remove extremely poor-quality images, some noise remains due to the inherent limitations of open-source data. Nevertheless, performance trends on CityLens-Mapillary closely mirror those on the original dataset, demonstrating that open-source platforms can still provide strong utility for socioeconomic prediction. These findings highlight the viability and promise of open-source alternatives like, especially for future research in urban socioeconomic sensing.

## A.4 NONLINEAR REGRESSOR RESULTS UNDER FEATURE-BASED REGRESSION SETTING

We further conduct additional experiments using nonlinear regressors, including Random Forest, XGBoost, and MLP, to regress on the 13-attribute vectors extracted from Gemma3-27B and Llama4-

Table 3: Number of valid prediction tasks per indicator across regions in CityLens vs. CityLens-Mapillary.

| Task | Google/Baidu | Mapillary |
|------|--------------|-----------|
| GDP | 1000 | 805 |
| Population | 1000 | 824 |
| House Price | 777 | 527 |
| Public Transport | 500 | 409 |
| Drive Ratio | 500 | 413 |
| Mental Health | 500 | 416 |
| Accessibility to Health | 1000 | 809 |
| Life Expectancy | 193 | 89 |
| Building Height | 1000 | 813 |
| Violent Crime | 396 | 345 |
| Bachelor Ratio | 500 | 382 |

Table 4: Main Results on Feature-Based Regression method from Mapillary data. The values in the table represent $R^2$ scores. "Mean" denotes the average performance across tasks, and "SD" refers to the standard deviation. In each row, bold indicates the best result, and underline denotes the second-best.

| Domain Tasks | GDP | Econ. Pop. | HP | Crime VC | Trans. PT | DR | Env. BH | MH | Health AH | LE | Edu. BR | Overall Mean | SD |
|---|---|---|---|---|---|---|---|---|---|---|---|---|---|
| Gemma3-4B | 0.390 | 0.132 | 0.100 | 0.013 | 0.274 | 0.170 | 0.532 | 0.075 | 0.332 | 0.092 | 0.097 | 0.201 | 0.153 |
| Gemma3-12B | 0.453 | 0.148 | 0.146 | 0.020 | 0.348 | 0.242 | 0.573 | 0.039 | 0.342 | 0.174 | 0.061 | 0.231 | 0.171 |
| Gemma3-27B | 0.471 | 0.188 | 0.165 | 0.016 | 0.299 | 0.274 | 0.583 | 0.074 | 0.353 | **0.205** | 0.088 | 0.247 | 0.165 |
| Qwen2.5VL-3B | 0.406 | 0.142 | 0.206 | 0.007 | 0.228 | 0.199 | 0.563 | 0.028 | 0.365 | 0.126 | 0.040 | 0.210 | 0.166 |
| Qwen2.5VL-7B | 0.401 | 0.161 | 0.150 | **0.056** | 0.376 | 0.205 | 0.554 | -0.189 | 0.357 | 0.043 | 0.160 | 0.207 | 0.196 |
| Qwen2.5VL-32B | 0.408 | 0.148 | **0.220** | -0.018 | 0.329 | 0.220 | 0.563 | -0.044 | 0.341 | 0.085 | 0.107 | 0.214 | 0.176 |
| Llama4-Scout | 0.401 | 0.161 | 0.053 | -0.010 | 0.295 | 0.288 | 0.570 | 0.078 | 0.361 | -0.0004 | **0.167** | 0.215 | 0.176 |
| Llama4-Maverick | 0.470 | 0.171 | 0.157 | -0.007 | 0.287 | 0.362 | 0.594 | 0.082 | 0.357 | -0.0004 | 0.088 | 0.233 | 0.188 |
| Mistral-small-3.1-24B | 0.462 | 0.175 | 0.109 | 0.021 | 0.376 | 0.281 | 0.560 | **0.112** | 0.366 | -0.0004 | 0.099 | 0.233 | 0.179 |
| Phi-4-multimodal | 0.438 | 0.170 | 0.138 | 0.017 | 0.308 | 0.240 | 0.576 | 0.034 | 0.286 | 0.133 | 0.064 | 0.219 | 0.166 |
| Nova-lite-v1 | 0.424 | 0.131 | 0.163 | 0.002 | 0.273 | 0.229 | 0.533 | 0.046 | 0.292 | -0.0004 | 0.116 | 0.201 | 0.163 |
| Minimax-01 | 0.433 | 0.139 | 0.206 | -0.016 | 0.362 | 0.250 | 0.587 | 0.030 | 0.374 | -0.0004 | 0.128 | 0.227 | 0.186 |
| Gemini-2.0-Flash | 0.443 | 0.157 | 0.130 | -0.002 | **0.444** | 0.326 | **0.607** | 0.093 | 0.357 | 0.059 | 0.111 | **0.248** | 0.187 |
| Gemini-2.5-Flash | 0.452 | **0.210** | 0.198 | 0.040 | 0.362 | 0.246 | 0.598 | 0.004 | 0.297 | -0.0004 | 0.085 | 0.227 | 0.183 |
| GPT-4o-mini | 0.394 | 0.135 | 0.151 | 0.007 | 0.273 | **0.371** | 0.533 | 0.071 | **0.384** | -0.0004 | 0.071 | 0.217 | 0.173 |
| GPT-4.1-mini | **0.478** | 0.185 | 0.092 | 0.019 | 0.350 | 0.259 | 0.574 | 0.011 | 0.304 | -0.0004 | 0.108 | 0.216 | 0.186 |
| GPT-4.1-nano | 0.429 | 0.179 | 0.037 | 0.005 | 0.347 | 0.227 | 0.520 | **0.112** | 0.332 | 0.025 | 0.113 | 0.211 | 0.166 |

Maverick. The results are summarized in the Table 5. While nonlinear regressors offer modest improvements over linear models in certain tasks such as GDP prediction, they do not consistently outperform across tasks. In particularly challenging indicators like Violent Crime and Mental Health, R² scores remain low or even negative for all nonlinear regressors, suggesting that the primary bottleneck is not the regression capacity, but rather a limitation in the expressiveness of the 13-attribute vectors extracted by current LVLMs.

Table 5: Results of nonlinear regressors, including Random Forest, XGBoost, and MLP.

| Method | Model | GDP | Pop. | HP | VC | PT | DR | BH | MH | AH | LE | BR |
|---|---|---|---|---|---|---|---|---|---|---|---|---|
| RF | Gemma3-27B | 0.563 | 0.234 | 0.106 | -0.335 | 0.477 | 0.587 | 0.618 | 0.181 | 0.302 | 0.240 | 0.256 |
| RF | Llama4-Scout | 0.516 | 0.239 | 0.114 | -0.063 | 0.443 | 0.448 | 0.575 | 0.207 | 0.304 | 0.176 | 0.266 |
| XGBoost | Gemma3-27B | 0.507 | 0.263 | 0.130 | -0.533 | 0.357 | 0.392 | 0.571 | 0.199 | 0.244 | 0.129 | 0.174 |
| XGBoost | Llama4-Scout | 0.549 | 0.153 | 0.210 | -0.072 | 0.411 | 0.370 | 0.556 | 0.180 | 0.294 | 0.112 | 0.161 |
| MLP | Gemma3-27B | 0.467 | 0.332 | 0.116 | 0.085 | 0.507 | 0.486 | 0.599 | -0.053 | 0.296 | -10.632 | 0.218 |
| MLP | Llama4-Scout | 0.413 | 0.247 | 0.192 | 0.005 | 0.317 | 0.402 | 0.610 | -0.155 | 0.309 | -1.042 | 0.102 |

A.5 Details about CityLens Dataset

A.5.1 Summary of Indicators in CityLens

We provide an overview of all indicators considered in the construction of the CityLens benchmark. Table 6 lists all 28 collected indicators, along with their data sources and whether they are ultimately selected as prediction tasks.

**Economy** Under the economy domain, we cover 7 critical indicators: GDP, house price, population, median household income, poverty 100%, poverty 200% and income Gini coefficient. For GDP, we utilize a global dataset that provides GDP estimates with a spatial resolution of 1 km Wang & Sun (2022). For population, we adopt estimates from WorldPop Tatem (2017), a global demographic dataset with 1 km spatial resolution that provides consistent population counts across countries. For house price, we collect data from multiple sources tailored to each country's context: (1) For US cities, we use the Zillow Home Value Index (ZHVI) (Zillow, 2020), available at the ZIP code level. We map these values to census tract boundaries using spatial overlays, enabling fine-grained local prediction. (2) For UK cities, we target the Middle Layer Super Output Area (MSOA) level, obtaining house price data from Han et al. (2023). (3) For Chinese cities, we collect house price data from LianJia (Platform, 2020), one of China's largest online real estate platforms. For median household income, poverty 100%, and poverty 200%, we obtain the raw data from SafeGraph. We obtain the ground-truth values for the income Gini coefficient from Zhang et al. (2025b).

**Transport** In the transport domain, we include seven indicators: PMT, VMT, PTRP, VTRP, walk and bike ratio, public transport ratio, and drive ratio, following the design of Fan et al. (2023). The underlying data is sourced from Survey (2017), which provides commuting behavior statistics at the census tract level across the United States. Table 7 outlines the definitions of the seven transport related indicators.

**Crime** In the crime domain, we focus on two indicators: violent crime incidence and non-violent crime incidence, both defined as the number of crime occurrences per census tract. The data is collected from the official websites of individual US cities (Chicago, 2019; New York, 2019; San Francisco, 2019), which publish annual crime reports and geolocated incident-level data.

**Health** For the health domain, we include 9 kinds of indicators to capture different dimensions of urban health outcomes: obesity, diabetes, cancer, no leisre-time physical activity (LPA), mental health, physical health, depression rate, accessibility to healthcare, and life expectancy. The first 7 tasks focus on the United States only, using data from "Local Data for Better Health" (PLACES). The Accessibility to Healthcare task is defined globally, using a dataset that quantifies walking-only travel time to the nearest healthcare facility (Weiss et al., 2020). The Life Expectancy task targets the United Kingdom, where we use data from Han et al. (2023) to obtain male life expectancy estimates at the MSOA level.

**Environment** In the environment domain, we consider two indicators: Carbon Emissions and Building Height. For carbon, we use global estimates from Oda & Maksyutov (2015). We use global building height data obtained from Pesaresi & Politis (2022), which provides global coverage at a spatial resolution of 100 meters.

**Education** Following Liu et al. (2023b), we use the Bachelor Ratio, defined as the proportion of residents holding a bachelor's degree or higher, as the target variable in the education domain. The ground-truth data for this indicator is obtained from SafeGraph (SafeGraph), which provides fine-grained demographic datasets across the United States.

A.5.2 Data Mapping and Aggregating

In CityLens, each region serves as a unit of prediction and is represented by 1 satellite image and 10 street view images. For each region, we associate one scalar label for the target indicator by mapping and aggregating raw tabular data from heterogeneous sources. Regarding the label mapping and aggregation strategies:

Table 6: Summary of 28 indicators in CityLens.

| Domain | Indicator | Source | Selected |
|---|---|---|---|
| Economy | GDP | Wang & Sun (2022) | ✓ |
| | House Price | Zillow (2020); Han et al. (2023) Platform (2020) | ✓ |
| | Population | Tatem (2017) | ✓ |
| | Median Income | SafeGraph | |
| | Poverty 100% | SafeGraph | |
| | Poverty 200% | SafeGraph | |
| | Income Gini Coefficient | Zhang et al. (2025b) | |
| Education | Bachelor Ratio | SafeGraph | ✓ |
| Crime | Violent | Chicago (2019); New York (2019) San Francisco (2019) | ✓ |
| | Non-Violent | Chicago (2019); New York (2019) San Francisco (2019) | |
| Transport | PMT | Survey (2017) | |
| | VMT | Survey (2017) | |
| | PTRP | Survey (2017) | |
| | VTRP | Survey (2017) | |
| | Walk and Bike | Survey (2017) | |
| | Drive Ratio | Survey (2017) | ✓ |
| | Public Transport | Survey (2017) | ✓ |
| Health | Obesity | PLACES | |
| | Diabetes | PLACES | |
| | LPA | PLACES | |
| | Cancer | PLACES | |
| | Mental Health | PLACES | ✓ |
| | Physical Health | PLACES | |
| | Depression Rate | PLACES | |
| | Life Expectancy | Han et al. (2023) | ✓ |
| | Accessibility to Healthcare | Weiss et al. (2020) | ✓ |
| Environment | Carbon Emissions | Oda & Maksyutov (2015) | |
| | Building Height | Pesaresi & Politis (2022) | ✓ |

Table 7: Definitions of the seven indicators in the Transport domain.

| Topic | Indicator | Label |
|---|---|---|
| Transport | %Population(>16)commute by driving alone | Drive Ratio |
| | Estimated personal miles traveled on a working weekday | PMT |
| | Estimated personal trips traveled on a working weekday | PTRP |
| | Estimated vehicle miles traveled on a working weekday | VMT |
| | Estimated vehicle trips traveled on a working weekday | VTRP |
| | %Population(>16)commute by public transit | Public Transit |
| | %Population(>16)commute by walking and biking | Walk and Bike |

- For Public Transport Ratio, Drive Ratio, Mental Health, and Violent Crime, we follow the methodology in Fan et al. (2023). The data is provided at the CT level, so we use CT as the key to directly map indicators to regions. For Bachelor Ratio, the original data is also available at the CT level, so we apply the same CT-based mapping.
- For House Price:
  - In the U.S., the original values are provided by Zillow at the ZIP code level. Using the official crosswalk between ZIP codes and CTs, we assign ZIP-level values to CTs. Since some CTs and ZIP codes do not align perfectly, we use averaging in cases of overlap.
  - In China, the raw dataset consists of individual records with latitude and longitude. We aggregate values by averaging all records falling within each satellite image's coverage.

- – In the U.K., the data is already available at the MSOA level, so we perform direct MSOA-to-region mapping.
- For Life Expectancy, we use the same MSOA-based strategy.
- For globally available GDP, Population, Building Height, and Accessibility to Healthcare, the original data is provided in GeoTIFF format. For each satellite image, we extract the values covering the image's geographic extent and compute the average as the region-level indicator.

## A.6 ADDITIONAL INFORMATION ABOUT THREE EVALUATION METHODOLOGIES

### A.6.1 PROMPT DESIGN AND CASE ANALYSIS

We provide additional insights into the design and behavior of LVLMs under the three evaluation methodologies. Table 8 summarizes the representative prompts used in the three paradigms, highlighting their structural differences and role-setting strategies. Figure 9 shows a case in the Direct Metric Prediction setting, where the model refuses to estimate GDP due to insufficient information. Figure 10 shows an example under the Normalized Estimation setting, where the model is asked to predict the bachelor ratio based on regional images. Figure 11 presents an example from the Feature-Based Regression method, where the model scores a street view image along 13 predefined urban visual attributes to support downstream prediction.

Table 8: Prompt comparison across the three evaluation methodologies in transport domain tasks.

| Method | Prompt |
|---|---|
| Direct Metric Prediction | Suppose you are a professional transport data analyst in {city}, {country}. Based on the provided satellite imagery and several street view photos, please estimate 'the {indicator}' in the census tract where these images are taken. Consider factors such as road infrastructure, visible traffic patterns, availability of public transport options, pedestrian walkways, and any other relevant details that might influence these transport behaviors in the area.
Please provide a single specific number (not a range or approximate value) for '{indicator}'. No explanation is needed.Example answer: {example num}. |
| Normalized Metric Estimation | Suppose you are a professional transport data analyst in {city}, {country}. Based on the provided satellite imagery and several street view photos, please estimate 'the {indicator}' in the census tract where these images are taken. Consider factors such as road infrastructure, visible traffic patterns, availability of public transport options, pedestrian walkways, and any other relevant details that might influence these transport behaviors in the area.
Please provide a single specific number for '{indicator}' (on a scale from 0.0 to 9.9). No explanation is needed. Example answer: 8.8. |
| Feature-Based Regression | Analyze the provided street view image. For each of the following 13 indicators, provide a score from 0.0 to 9.9 representing its presence or prominence in the image. The output should only be the indicator name followed by its score, one indicator per line. No need for explanations or additional text.
Indicators: Person; Bike; Heavy Vehicle; Light Vehicle; Façade; Window & Opening; Road; Sidewalk; Street Furniture; Greenery - Tree; Greenery - Grass & Shrubs; Sky; Nature
Example: Person: 2.5; Bike: 0.0; ...... |

### A.6.2 SUPPLEMENTARY COMPARISON OF THREE EVALUATION METHODOLOGIES

We present two supplementary figures in Figure 12: one comparing Feature-Based Regression with Direct Metric Prediction, and the other with Normalized Estimation. These visualizations enable side-by-side assessment of performance across all socioeconomic indicators.

Figure 9: An example of model refusal in the Direct Metric Prediction setting.

Figure 10: Case example for Normalized Metric Estimation.

From the figures, it is evident that the Feature-Based approach, where the large vision-language model acts as a feature enhancer, consistently outperforms the Direct and Normalized approaches, in which the model is expected to behave as a numerical predictor. This suggests that current LVLMs, while powerful in perceptual and language tasks, are still more effective when used to extract structured visual representations rather than to directly generate precise socioeconomic estimates. Although large vision-language models have made impressive strides, accurately predicting fine-grained, region-level socioeconomic indicators remains highly challenging, highlighting the need for further advancements.

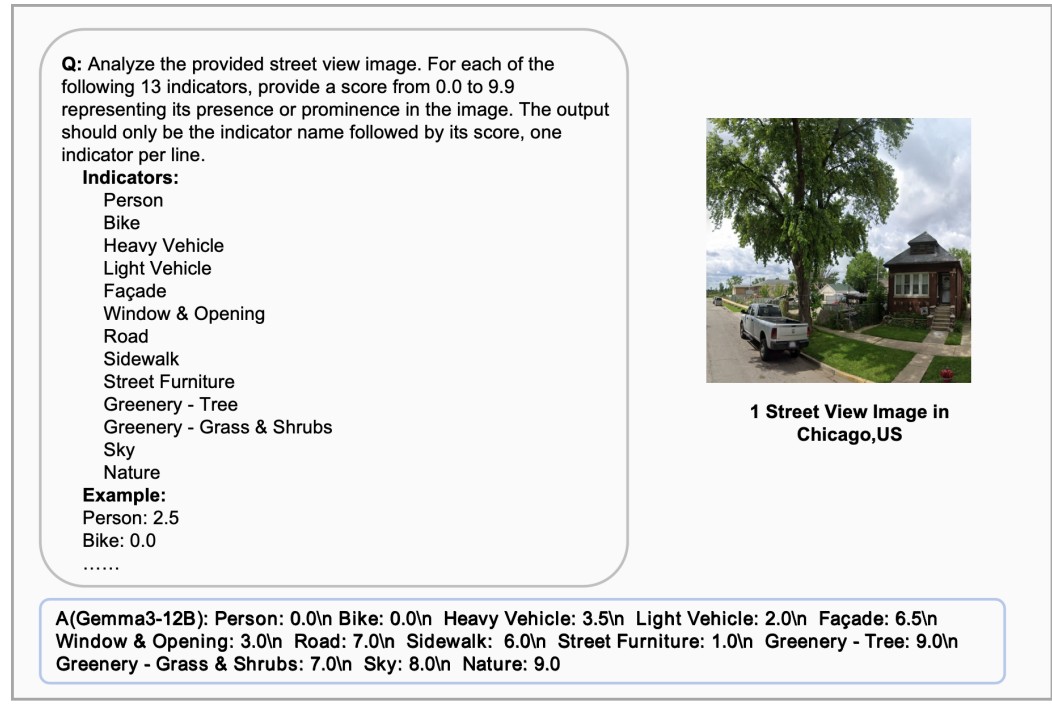

Figure 11: Prompt template for guiding large vision-language models to extract 13 visual features from a street view image.

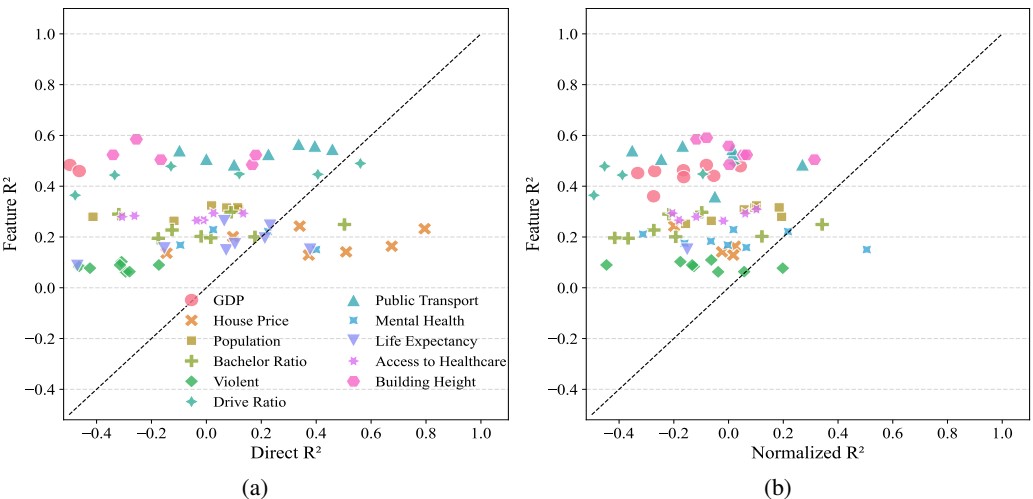

(a)                                                                      (b)

Figure 12: (a) shows the performance comparison between Feature-Based Regression and Direct Metric Prediction. (b) compares Feature-Based Regression with Normalized Estimation..

## A.7   ADDITIONAL EXPERIMENTAL SETUP DETAILS

### A.7.1   LVLMS

We consider a diverse set of LVLMs as baselines to benchmark our proposed methods. The selected models include both open-source and proprietary systems, covering a range of model sizes and capabilities. We choose Gemma3-4B/12B/27B (Team, 2025), Qwen2.5VL-3B/7B/32B (Bai et al., 2025), Llama4-Scout/Maverick (AI, 2025a), Mistral-small-3.1-24B (AI, 2025b), Phi-4-multimodal (Aboue-lenin et al., 2025), MiniMax-01 (Li et al., 2025), Gemini-2.0-flash/Gemini-2.5-flash (DeepMind, 2025), GPT-4o-mini (Achiam et al., 2023), GPT-4.1-mini/nano (OpenAI, 2025) and Amazon-Nova-

Lite (Amazon, 2025). One thing to note is that models in the Gemini series can accept at most 10 images as input. Therefore, for this series, we use 1 satellite image and 9 street view images per region to stay within the model's input constraints.

### A.7.2 METRICS

For evaluation, we adopt two commonly used metrics in socioeconomic prediction tasks: coefficient of determination ($R^2$) and normalized root mean squared error (nRMSE). Higher $R^2$ indicates better performance, with 1.0 representing perfect prediction. Lower nRMSE values indicate more accurate predictions.

### A.7.3 CHOICE OF TASK INPUT

While prior work such as Fan et al. (2023) uses 20 or more street view images to represent each region, we find that this setup is often impractical for LVLMs. Specifically, we initially experiment with 20 street view images per region, but observe that this would significantly increase the computational cost, and exceed the input limits of models like Gemini, which can only process up to 10 images per inference, and also frequently hit the token length limit of other models. Therefore, we adopt a compromise of 10 images per region to ensure compatibility across models while maintaining sufficient visual context.

### A.7.4 THE EXAMPLE OF CoT PROMPTING

Following the designs of Zhang et al. (2025b) and Xu et al. (2024), we implement a Chain-of-Thought (CoT) prompting strategy tailored to the urban socioeconomic sensing context. Here, we present an example CoT prompt designed specifically for the House Price task.

---

**Q:** Suppose you are a professional real estate appraisal expert in Leeds, United Kingdom. Based on the provided satellite imagery and several street view photos, please estimate 'house price' in the msoa area where the images are taken. Consider factors such as location, visible property features, neighborhood condition, and any other relevant details.
Satellite Image: <image>
Street View Images: <image> <image> <image> <image> <image> <image> <image> <image> <image> <image>

To perform better on this task, please answer by adopting a step-by-step reasoning approach:
Step1 <Summary>:
Explain your overall strategy for estimating the house price based on the given satellite and street view images. Describe how you will approach the task using visual evidence, and mention the types of features you plan to focus on.
Step2 <Caption>:
Next, analyze each of the provided street view and satellite images. Clearly list the specific visual features you observe that might affect house prices, then summarize these into an overall socioeconomic profile of the area.
Step 3 <Calculation>:
Based on your previous analysis, determine the approximate house-price level for this area and briefly explain the primary reasons behind your choice.
Step4 <Answer>:
Output a single number only representing the estimated average house price for the area in the format of <answer>NUMBER</answer>.

---

Figure 13: CoT prompt example for the House Price task.

### A.8 ALTERNATIVE OUTPUT DESIGNS AND EVALUATION STRATEGIES

### A.8.1 EXPLORING MULTIPLE-CHOICE STYLE ANSWERING FORMATS

All three evaluation formats used in CityLens can be naturally framed as regression tasks. This design choice aligns with established paradigms in recent related works Manvi et al. (2024b), Manvi et al. (2024a) and Zhang et al. (2025b), where urban socioeconomic indicators are typically formulated as continuous variables, and numerical prediction remains the primary objective.

We acknowledge that multiple-choice formats can be useful for probing certain model capabilities, such as semantic understanding or categorical reasoning. During the development of CityLens, we explore this possibility by designing a multiple-choice version of the Population prediction task. To generate negative choices, we adopt a simple yet controlled strategy: for each ground-truth population value, three distractors were sampled from a fixed pool of plausible but incorrect values—for example,

0.05, 20, and 300. Below is an example we tested using three models. Preliminary results suggest that while the multiple-choice setting reduces the output space and improves answer interpretability, it also significantly lowers task difficulty compared to free-form regression. Consequently, we ultimately opted to focus on open-ended, regression-based evaluation, which we believe more accurately reflects the complexity and realism of urban socioeconomic sensing.

This decision is supported by emerging literature in LLM evaluation. Prior studies Aidar Myrza-khan (2024) Li et al. (2024a) have identified several systematic limitations in multiple-choice-based evaluations, including selection bias, position sensitivity, and a tendency toward random guess-ing—especially in smaller models. These issues may lead to inflated estimates of model capability and fail to capture the inherent complexity of the task. Moreover, the multi-choice format may fail to capture the nuanced reasoning or visual understanding required for tasks like urban socioeconomic sensing. In contrast, free-form numerical prediction—despite being more challenging—offers a more direct and faithful reflection of a model's ability to process multimodal information and generate semantically grounded outputs.

### A.8.2 STREET VIEW CAPTION EMBEDDING FOR SOCIOECONOMIC REGRESSION

We conduct an exploratory experiment inspired by prompt design strategies from UrbanVLP Hao et al. (2025), applying them to generate captions for street view images. As illustrated in Figure 14a, each image is accompanied by contextual information including the city name, geographic coordinates, and scores across 13 predefined visual features. This multimodal input is then fed into a large vision-language model to produce a descriptive caption of the scene.

To leverage the semantic richness of these captions, we pass the generated texts through a BERT encoder to obtain fixed-length embeddings. These embeddings are subsequently used as input features for downstream regression tasks targeting urban socioeconomic indicators. As shown in Figure 14b, this caption-based embedding approach achieves the best performance on the population prediction task, outperforming all other methods.

Q: Analyze the image of streetview in {city} in a comprehensive and detailed manner: The coordinate of the streetview image is {Longitude}, {Latitude}. The visual feature score of the streetview image is Facade: {score}. Road: {score}. Greenery-Grass & Shrubs: {score}. Greenery-Tree: {score}. Street Furniture: {score}. Person: {score}. Bike: {score}. Heavy Vehicle: {score}. Light Vehicle: {score}. Window & Opening: {score}. Sidewalk: {score}. Sky: {score}. Nature: {score}.

| Method | Population $R^2$ |
|---|---|
| Direct | $< -0.5$ |
| Normalized | -0.1570 |
| Feature-Based | 0.2518 |
| Caption-Embed | 0.3498 |

(a)                                    (b)

Figure 14: (a) Prompt for Caption-Embed Method. (b) Population $R^2$ values for different methods.

### A.9 BIAS AUDITS

In CityLens, one notable issue is the underrepresentation of Global South cities in some tasks, which may pose a risk of reinforcing biases in model predictions. To clarify, for several tasks—such as Public Transport Ratio and Mental Health—Global South cities are excluded due to the unavailability of high-quality ground-truth indicators in these regions.

Building on the analysis presented in Section 3.3, we conduct a preliminary bias audit on the GDP prediction task. Specifically, we use the Global North and Global South classification provided by Wikipedia to categorize cities as follows:

- Global North: San Francisco, New York, Tokyo, London, Paris, Sydney
- Global South: Beijing, Shanghai, Mumbai, Moscow[1], Sao Paulo, Nairobi, Cape Town

We then compare model performance on the GDP prediction task between these two groups. As shown in the table below, we observe that models perform significantly better on Global North cities,

---

[1]Note: the classification of Moscow is debated in some literature.

achieving substantially higher $R^2$ and lower nRMSE, suggesting better predictive reliability and fit. This performance gap highlights a geographic disparity in model behavior and may point to underlying biases in how current LVLMs generalize across different socioeconomic and cultural contexts. The fact that these differences emerge under a consistent task formulation and input structure strengthens the case for further bias auditing and fairness-aware model evaluation. Additionally, in Figure 5(a), we observe that Mumbai and Moscow yield significantly negative $R^2$ values, further reinforcing the presence of geographic bias in model performance.

Table 9: $R^2$ and nRMSE for Global North and Global South Cities

| City | $R^2$ | nRMSE |
|------|-------|-------|
| Global North | 0.399 | 0.787 |
| Global South | 0.168 | 0.869 |

### A.10 HYPOTHESES ON LVLMS FOR URBAN SOCIOECONOMIC SENSING

We propose to view visual concepts along two dimensions: low-level and high-level.

- Low-level concepts refer to basic, concrete features such as greenery, Street Furniture, or vehicle density, which are often directly observable and recognizable by traditional machine learning models.

- In contrast, high-level concepts—such as signs of poverty, infrastructure quality, or commercial activity—are abstract constructs that emerge from combinations of multiple visual and contextual signals. These are typically more entangled with socioeconomic meaning and are closer to human-level interpretation.

One of the key strengths of LVLMs lies in their ability to go beyond recognizing low-level features and implicitly perceive and reason about high-level visual semantics. While this ability represents a significant advancement, it also introduces major challenges—it becomes extremely difficult to analyze whether and how the model correctly interprets high-level visual features. For instance, when it comes to abstract indicators like Bachelor Ratio, it is inherently difficult to isolate a single visual factor as the dominant predictor. These outcomes are typically influenced by acombination of visual signals (e.g., signs of poverty, commercial activity, infrastructure conditions), and the relationship between them is complex and entangled.

While these models exhibit impressive capabilities in perceiving a broader spectrum of visual concepts, they also face well-known limitations, most notably hallucination and instability in perception. When identifying low-level features, LVLMs may overlook relevant signals or generate hallucinated content. At the high-level, models may fail to correctly identify or interpret key contextual cues related to the target indicator, leading to incomplete or biased reasoning. For example, we observe in the BR task that GLM-4.1v-Thinking tends to over-reason, gradually drifting away from task-relevant semantics.

The above discussion is grounded in our hypothesis about how LVLMs operate in urban socioeconomic sensing tasks: When presented with a socioeconomic prediction task, an LVLM may follow a multi-level reasoning process. It first identifies high-level visual concepts that are semantically associated with the target indicator. For example, to estimate the bachelor ratio, the model may consider abstract cues such as signs of poverty, commercial activity, and various other concepts. To support the recognition of these high-level visual concepts, the model must detect corresponding low-level visual features from the input images—such as building density, greenery coverage, and so on. These low-level features are more concrete and directly extractable from satellite or street view images. The model then aggregates these features into high-level visual concepts, which are further cross-checked and composed through reasoning to produce the final prediction.

However, if key low-level features are missed or hallucinated during recognition, the resulting high-level concept understanding may be distorted, ultimately leading to inaccurate predictions. Therefore, we believe that understanding why LVLMs struggle with urban socioeconomic sensing is a highly meaningful and non-trivial challenge that deserves deeper investigation. Moreover, the validity of our proposed hypothesis regarding the multi-level visual reasoning process in LVLMs is itself an open question.

