# OpenReview forum: "CityLens: Evaluating Large Vision-Language Models for Urban Socioeconomic Sensing"
_ICLR.cc/2026/Conference — ICLR 2026 Poster_

### Official Review · Reviewer_Zxk9 · 2025-10-20

**Soundness:** 2
**Presentation:** 3
**Contribution:** 1
**Rating:** 2
**Confidence:** 4

**Summary:**

This paper introduces CityLens, which is a benchmark for evaluating the capabilities of LVLM in urban socioeconomic deduction. The benchmark is comprehensive acorss 11 prediction tasks across 6 domains e.g., economy and health. The authors evaluate 17 SOTA LVLMs using three evaluation protocols. The primary findings are that current LVLMs struggle significantly with these tasks and that the feature regression method outperforms the direct and zero-shot predictions.

**Strengths:**

The primary strength of this paper is the creation of a large-scale dataset for this urban tasks. The effort in collecting, aligning, and processing satellite and street view images from multiple sources, and 11 distinct socioeconomic indicators is non-trivial. The dataset could be of use for other research efforts.
The reproducibility is very good. The authors provide an alternative version of the dataset using only publicly accessible Mapillary images (CityLens-Mapillary), which is a commendable step towards transparency and reproducibility.

**Weaknesses:**

1.This paper's central contribution is a new benchmark rather than a new method. The models and evaluation techniques are nothing new. This is not in line with the scope of ICLR

2The main takeaway is that current LVLMs perform not good on this complex task with many R2 scores near 0. I admit "negative results" are valuable, the paper offers limited deep insight into the reasons behindwhy these representations fail.

3The finding that COT is task-dependent is already a well-established consensus in the broader LLM community. The claim that specific vision encoders CLIP outperform others is also expected.

4.The comparison is not fair. Feature based methods have a LASSO regressor while other evaluation protocols do not have.

5.The finding that satellite imagery has "minimal impact" is based on an unbalanced comparison of 1 satellite image vs. 10 street view images. It is unsurprising that the information from 10 ground-level images dominates the information from one overhead image.

**Questions:**

What would happen if you only use 1 satellite image and compare it with 10 street ones?

---

> ### Author Response · Authors · 2025-11-21
> **Author Response to Reviewer Zxk9 (part 1/3)**
>
> Thank you for your thoughtful comments and for taking the time to review our paper. Below, we have included detailed responses to your feedback. In the revised version of the manuscript, we have updated the paper accordingly and highlighted all changes in blue for your convenience. We hope our replies sufficiently address your concerns and enhance clarity. Please do not hesitate to reach out if you have any further questions or feedback.
>
> > **For W1: Scope concern**
>
> We respectfully clarify that our submission is indeed in line with ICLR's scope, particularly under the primary subject area of **"datasets and benchmarks".** In this work, we present CityLens, a large-scale multimodal benchmark covering 17 globally distributed cities across 6 key urban domains. As part of this effort, we also release a dataset, where we collect, process, and match socioeconomic indicators with urban imagery from various original sources. The data used in the benchmark is a subset of this dataset. We define 11 prediction tasks, introduce 3 evaluation paradigms, and benchmark 17 state-of-the-art LVLMs, making CityLens the most comprehensive benchmark to date for urban socioeconomic sensing. It offers a unified framework to evaluate and understand the capabilities and limitations of LVLMs in real-world, socially impactful domains.
>
> In fact, **several papers centered on benchmarks and datasets were accepted at ICLR 2025**, including: GlycanML [1], CEB [2], LiveBench [3], Dysca [4] and so on. Like these works, our paper does not propose new modeling techniques, but instead makes methodologically meaningful contributions by systematically evaluating model capabilities in an underexplored yet socially important area of urban socioeconomic sensing. To this end, we construct the most geographically and semantically diverse benchmark to date. We believe CityLens opens a valuable new testing ground for the community and aligns closely with ICLR’s scope.
>
> Furthermore, we note that **the other three reviewers all gave positive assessments (score: 6) and none raised concerns regarding the paper's alignment with ICLR's scope**. We sincerely hope that the reviewer might reconsider this issue in light of the broader context and contribution of our work.
>
> [1] Xu, Minghao, et al. "GlycanML: A Multi-Task and Multi-Structure Benchmark for Glycan Machine Learning." ICLR 2025
>
> [2] Wang, Song, et al. "CEB: Compositional Evaluation Benchmark for Fairness in Large Language Models." ICLR 2025
>
> [3] Colin White, et al. "LiveBench: A Challenging, Contamination-Limited LLM Benchmark." ICLR 2025
>
> [4] Zhang, Jie, et al. "Dysca: A Dynamic and Scalable Benchmark for Evaluating Perception Ability of LVLMs." ICLR 2025

---

> ### Author Response · Authors · 2025-11-21
> **Author Response to Reviewer Zxk9 (part 2/3)**
>
> > **For W2: Error Cases of LVLMs on Challenging tasks**
>
> We thank the reviewer for this insightful suggestion, which helps improve the completeness of our paper. In response, and also incorporating related feedback from Reviewer gfN8, we have added a new paragraph in "**Section 3.5: Error Analysis and Upper Bounds of LVLMs**" to qualitatively analyze failure cases on challenging tasks such as Mental Health. The corresponding text is also included below. (Due to format constraints, figures are not shown here but are included in the paper.)
>
> `Error Cases on Challenging tasks` To better understand why LVLMs underperform on challenging tasks, we analyze representative errors observed under both the Feature-Based Regression and CoT Prompting setups. As illustrated in Figure 8, errors can arise from both visual perception and linguistic reasoning. For example, during feature extraction, Gemma-3-4B fails to detect small but meaningful elements such as street signs, hallucinates non-existent persons, and underestimates visible greenery, assigning a low score to "Grass \& Shrubs". These errors reveal a lack of fine-grained visual grounding and semantic alignment, which can propagate into downstream reasoning and prediction. We also observe reasoning errors in the CoT setting. For instance, in a Mental Health prediction case, the model overly focuses on a few old and modest houses while overlooking numerous well-maintained, even upscale beachfront apartments visible in the street view images. Moreover, it fails to leverage the region's proximity to water, which is a known factor with strong aesthetic and calming effects that are directly linked to mental well-being. This suggests that current LVLMs may struggle to appropriately weigh holistic environmental cues during reasoning.
>
> > **For W3: Response to comment on CoT and vision encoder findings**
>
> `CoT prompting.`  It is commonly believed that incorporating Chain-of-Thought (CoT) prompting can help solve complex tasks, including challenging inequality indicators, by providing a clear reasoning path [1, 2]. Our contribution lies in showing that, in the context of urban socioeconomic sensing, the effectiveness of CoT prompting varies significantly across both tasks and models. We systematically identify which tasks (like house price) tend to benefit from CoT, and which tasks (like drive ratio) tend to suffer from it. Similarly, we report which models perform better or worse when CoT is applied, and offer plausible explanations for these trends based on the nature of the tasks and the model behaviors.
>
> `Vision encoder.` We respectfully note that **the choice and evaluation of vision encoders remains an active area of research**, with many recent studies exploring how different pretraining objectives and modality configurations affect downstream performance across diverse tasks [3, 4]. These works have shown that **no single vision encoder universally dominates**; rather, performance is task-dependent, particularly when comparing encoders such as CLIP and SigLIP. In this context, our contribution lies in empirically identifying that **CLIP performs best in the specific setting of extracting features from street view imagery** for urban socioeconomic sensing. We consider this finding to be non-trivial and context-dependent, offering important insights into the suitability of vision encoders for this underexplored yet socially relevant task. Notably, Reviewer gfN8 asks us to explain why CLIP performs best among the evaluated vision encoders, which further indicates that our findings are not self-evident.
>
> [1] Zhang, Yunke, et al. "Perceiving Urban Inequality from Imagery Using Visual Language Models with Chain-of-Thought Reasoning." WWW 2025
>
> [2] Wei, Jason, et al. "Chain-of-Thought Prompting Elicits Reasoning in Large Language Models." NeurIPS 2022
>
> [3] Tong, Shengbang, et al. "Cambrian-1: A Fully Open, Vision-Centric Exploration of Multimodal LLMs." NeurIPS 2024
>
> [4] Siddharth Karamcheti, et al. "Prismatic VLMs: Investigating the Design Space of Visually-Conditioned Language Models." ICML 2024

---

> ### Author Response · Authors · 2025-11-21
> **Author Response to Reviewer Zxk9 (part 3/3)**
>
> > **For W4: Response to concern about the fairness of evaluation protocols**
>
> We thank the reviewer for raising this concern. However, we would like to clarify that the three evaluation methodologies in our benchmark are designed to assess different aspects of LVLM capabilities. In the first two paradigms, Direct Metric Prediction and Normalized Estimation, we use the LVLM as a direct predictor, estimating socioeconomic indicators in an end-to-end manner. In contrast, the third paradigm, Feature-Based Regression, evaluates the LVLM’s ability to extract visual features from street view images, which are then fed into a downstream LASSO regressor for final prediction. These are fundamentally different evaluation setups, and our benchmark is explicitly designed to compare how LVLMs perform under each of these three distinct evaluation methodologies. Therefore, we believe the concern about fairness does not apply in this context.
>
> > **For W5 & Q1: Respective roles of street view and satellite imagery**
>
> We appreciate the reviewer's comment, but we respectfully disagree with the notion that comparing 1 satellite image to 10 street view images constitutes an unbalanced input setting, as this notion overlooks the fundamental differences in spatial scope and informational structure between the two modalities. Street view images capture fine-grained, semantically rich visual details that closely align with human perception and are often highly informative for socioeconomic prediction. However, they are also sparse and highly localized, typically covering only a few discrete points within a region. In contrast, a single satellite image provides a macroscopic overview of the entire area, encoding broader spatial patterns such as building density, urban form, and road network structure, features that are difficult to capture from street view imagery alone. One satellite image typically covers an area that would require thousands of street view images to represent equivalently in terms of spatial coverage.
>
> Moreover, as noted by Reviewer DkC7 referencing FlexiReg [1], satellite imagery has been found to be more discriminative than street view imagery for urban representation tasks, even under a setting where the satellite-to-street view ratio was as low as 1:64. This underscores that **information content cannot be directly equated to input quantity**. Therefore, the finding that 10 street view images may dominate 1 satellite image is not an obvious or trivial outcome within the context of applying LVLMs to urban socioeconomic prediction tasks.
>
> We extend the analysis in Figure 6b by conducting additional ablation experiments where we removed street view images and used only satellite imagery as input. The results are shown in the table below, and we have also updated **Section 3.3: Impact of Input Modalities** in the paper to include these new results. First, our results suggest that street view images provide more semantically rich and fine-grained visual cues such as building façades, commercial signage, and infrastructure quality. These ground-level features may be more tightly coupled with socioeconomic indicators and are also more interpretable to current LVLMs, which have been pretrained extensively on image–language pairs featuring such localized and human-centric content. Second, while satellite imagery performs relatively weaker, it still carries independent informational value. It offers spatial context such as urban morphology, building layout, and road networks, but may not provide the same level of semantic density at the resolution used in our benchmark.
>
>
> | Task        | Input Modality          | $R^2$ |
> | ----------- | ----------------------- | ----- |
> | House Price | Street view + Satellite | 0.373 |
> |             | Street view             | 0.374 |
> |             | Satellite               | 0.249 |
> | Drive       | Street view + Satellite | 0.561 |
> |             | Street view             | 0.567 |
> |             | Satellite               | 0.160 |
> | Public      | Street view + Satellite | 0.395 |
> |             | Street view             | 0.376 |
> |             | Satellite               | 0.277 |
>
> [1] Sun, Fengze, et al. "FlexiReg: Flexible Urban Region Representation Learning." KDD 2025

---

### Official Review · Reviewer_gfN8 · 2025-10-31

**Soundness:** 3
**Presentation:** 3
**Contribution:** 3
**Rating:** 6
**Confidence:** 3

**Summary:**

This paper proposes CityLens, a comprehensive benchmark dataset designed to evaluate the capabilities of Large Vision-Language Models (LVLMs) in predicting urban socioeconomic indicators from satellite and street view imagery. Specifically, this multimodal dataset covers 17 cities across 6 continents, 6 socioeconomic domains (economy, education, crime, transport, health, environment), and 11 specific prediction tasks. Building upon this, the authors systematically evaluate 17 state-of-the-art LVLMs using three distinct paradigms: Direct Metric Prediction, Normalized Metric Estimation, and Feature-Based Regression. The key findings reveal that while LVLMs show promise, they generally struggle with accurate and generalizable socioeconomic prediction, with performance varying significantly across tasks, models, and geographic locations.

**Strengths:**

1. The curated CityLens dataset is a major contribution to the community.
2. The benchmarking results are comprehensive and have a large coverage of state-of-the-art LVLMs.
3. The use of three evaluation paradigms (Direct, Normalized, Feature-Based) is technically sound.
4. The paper is well organized and easy to follow.

**Weaknesses:**

1. The evaluation is primarily zero-shot or few-shot. A natural question is how much these models could improve if fine-tuned on the CityLens dataset. While the authors mention this as a future direction, a preliminary fine-tuning experiment on one or two models would have strengthened the paper by establishing a potential performance upper bound.
2. While the paper diagnoses what models struggle with (e.g., mental health, life expectancy), a more detailed qualitative analysis of why could be beneficial. For instance, providing examples of specific visual cues that models misinterpret or fail to utilize for the most challenging tasks would offer deeper mechanistic insights.

**Questions:**

1. In the Feature-Based Regression paradigm, the authors show that CLIP-based encoders perform best. Was the satellite image used as an input to the LVLM during this feature extraction step, or was this based purely on the street view images? Could the superior performance of CLIP be attributed to its training on web-scale image-text pairs, which might include more diverse urban scenes?
2. Did you try nonlinear regressors for the 13-attribute vectors (RF/GBM/MLP)? If not, is the conclusion “LVLM features aren’t sufficient” or “the linear readout is underpowered”?

---

> ### Author Response · Authors · 2025-11-21
> **Author Response to Reviewer gfN8 (part 1/2)**
>
> We are encouraged that the reviewer finds our paper well-organized, the experimental results comprehensive, and the contributions valuable to the community. We also thank the reviewer for all the constructive feedback, which helped us revise our paper to strengthen it. In the revised version of the manuscript, we have updated the paper accordingly and highlighted all changes in blue for your convenience. We hope the following revisions and clarifications can address the reviewer's remaining concerns:
>
> > **For W1: Evaluation of fine-tuned LVLMs**
>
> Thank you for the insightful suggestion. We conduct preliminary supervised fine-tuning experiments on Qwen2.5-VL-7B, Qwen3-VL-8B, and Llama3.2-VL-11B using additional data from the CityLens dataset that is not included in the benchmark evaluation. The fine-tuning follow the 'Direct Metric Prediction' formulation, where the LVLM directly predicts socioeconomic indicators from visual input. It is important to note that the training data does not include samples for three indicators: House Price, Violent Crime, and Life Expectancy, so we exclude these tasks from its evaluation. The results of these additional experiments are summarized in the following table, and we have updated Table 2 in the latest version of the paper accordingly.
>
> General state-of-the-art LVLMs exhibit poor performance on many tasks in the CityLens benchmark, often achieving near-zero or even negative R² scores. In contrast, fine-tuned LVLMs, regardless of their base model or parameter size, **consistently demonstrate strong predictive capabilities across nearly all tasks**. These findings not only highlight the promising potential of LVLMs for urban socioeconomic sensing, but also provide a preliminary estimate of the performance upper bound that such models can achieve when appropriately adapted to this domain. This reinforces our central motivation for benchmarking LVLMs in this setting and underscores the importance of developing domain-specific LVLMs to address this real-world and socially significant challenge.
>
>
> | model name                 | GDP   | Pop.  | PT    | DR    | BH    | MH    | AH    | BR    |
> | -------------------------- | ----- | ----- | ----- | ----- | ----- | ----- | ----- | ----- |
> | Fine-tuned Qwen2.5-VL-7B   | 0.628 | 0.231 | 0.502 | 0.628 | 0.872 | 0.418 | 0.364 | 0.442 |
> | Fine-tuned Qwen3-VL-8B     | 0.626 | 0.107 | 0.545 | 0.638 | 0.869 | 0.397 | 0.304 | 0.536 |
> | Fine-tuned Llama3.2-VL-11B | 0.562 | 0.287 | 0.348 | 0.256 | 0.829 | 0.248 | 0.256 | 0.157 |
>
> > **For W2: Error Cases of LVLMs on Challenging tasks**
>
> We thank the reviewer for this insightful suggestion, which helps improve the completeness of our paper. In response, we have added a new paragraph in  **"Section 3.5: Error Analysis and Upper Bounds of LVLMs"** to qualitatively analyze failure cases on challenging tasks such as Mental Health. The corresponding text is also included below. (Due to format constraints, figures are not shown here but are included in the paper.)
>
> `Error Cases on Challenging tasks` To better understand why LVLMs underperform on challenging tasks, we analyze representative errors observed under both the Feature-Based Regression and CoT Prompting setups. As illustrated in Figure 8, errors can arise from both visual perception and linguistic reasoning. For example, during feature extraction, Gemma-3-4B fails to detect small but meaningful elements such as street signs, hallucinates non-existent persons, and underestimates visible greenery, assigning a low score to "Grass \& Shrubs". These errors reveal a lack of fine-grained visual grounding and semantic alignment, which can propagate into downstream reasoning and prediction. We also observe reasoning errors in the CoT setting. For instance, in a Mental Health prediction case, the model overly focuses on a few old and modest houses while overlooking numerous well-maintained, even upscale beachfront apartments visible in the street view images. Moreover, it fails to leverage the region's proximity to water, which is a known factor with strong aesthetic and calming effects that are directly linked to mental well-being. This suggests that current LVLMs may struggle to appropriately weigh holistic environmental cues during reasoning.

---

> ### Author Response · Authors · 2025-11-21
> **Author Response to Reviewer gfN8 (part 2/2)**
>
> > **For Q1: Explanation for CLIP's Superior Performance**
>
> We thank the reviewer for the thoughtful question. As described in Section 2.2, in the Feature-Based Regression setting, only street view images are used for feature extraction; satellite imagery and city names were not included as inputs. To make the task design of CityLens clearer, we have also added Figure 4 in this section, which illustrates prompt examples for each evaluation paradigm.
>
> We believe CLIP's superior performance over DINOv2 and IN1K stems primarily from differences in pretraining objectives and modality coverage. CLIP is trained with a vision-language contrastive objective, allowing it to align visual features with rich semantic information from language. In contrast, both DINOv2 and IN1K learn visual representations using only visual signals [1], which may limit their ability to capture high-level, abstract urban semantics. Additionally, CLIP's web-scale pretraining dataset includes internet-sourced images that may contain more diverse urban scenes and socioeconomic visual cues that are underrepresented or absent in datasets like ImageNet or those used for DINOv2 [2]. As for SigLIP, although it also employs a vision-language contrastive objective, its training data and loss formulation differ from those of CLIP. Since SigLIP’s pretraining corpus is not publicly disclosed, it is difficult to conduct a detailed analysis of its suitability for socioeconomic sensing tasks. Moreover, recent studies have shown that no single vision encoder consistently outperforms others across all tasks; performance is often task-dependent, particularly when comparing models like CLIP and SigLIP [1,2]. Our experiments provide empirical evidence that CLIP yields more effective representations than other vision encoders in the context of urban socioeconomic sensing, particularly when extracting features from street view imagery.
>
> (Note: IN1K refers to a standard Vision Transformer pretrained for classification on ImageNet-21K and then fine-tuned on ImageNet-1K.)
>
> [1] Tong, Shengbang, et al. "Cambrian-1: A Fully Open, Vision-Centric Exploration of Multimodal LLMs." NeurIPS 2024
>
> [2] Siddharth Karamcheti, et al. "Prismatic VLMs: Investigating the Design Space of Visually-Conditioned Language Models." ICML 2024
>
> > **For Q2: Results of nonlinear regressors**
>
> We thank the reviewer for this valuable suggestion. In response, we conduct additional experiments using nonlinear regressors, including Random Forest, XGBoost, and MLP, to regress on the 13-attribute vectors extracted from Gemma3-27B and Llama4-Maverick. The results are summarized in the table below. In the latest version of the paper, we have also added these new results to Appendix A.4.
>
> While nonlinear regressors offer modest improvements over linear models in certain tasks such as GDP prediction, they do not consistently outperform across tasks. In particularly challenging indicators like Violent Crime and Mental Health, R² scores remain low or even negative for all nonlinear regressors, suggesting that the primary bottleneck is not the regression capacity, but rather a limitation in the expressiveness of the 13-attribute vectors extracted by current LVLMs.
>
>
> | Method  | Model         | GDP   | Pop.  | HP    | VC     | PT    | DR    | BH    | MH     | AH    | LE      | BR    |
> | ------- | ------------- | ----- | ----- | ----- | ------ | ----- | ----- | ----- | ------ | ----- | ------- | ----- |
> | LASSO   | Gemma3-27B    | 0.463 | 0.324 | 0.141 | 0.077  | 0.567 | 0.525 | 0.591 | 0.211  | 0.283 | 0.245   | 0.297 |
> | LASSO   | Llama-4-Scout | 0.460 | 0.264 | 0.164 | 0.090  | 0.508 | 0.479 | 0.524 | 0.168  | 0.280 | 0.155   | 0.197 |
> | RF      | Gemma-3-27B   | 0.563 | 0.234 | 0.106 | -0.335 | 0.477 | 0.587 | 0.618 | 0.181  | 0.302 | 0.240   | 0.256 |
> | RF      | Llama-4-Scout | 0.516 | 0.239 | 0.114 | -0.063 | 0.443 | 0.448 | 0.575 | 0.207  | 0.304 | 0.176   | 0.266 |
> | XGBoost | Gemma-3-27B   | 0.507 | 0.263 | 0.130 | -0.533 | 0.357 | 0.392 | 0.571 | 0.199  | 0.244 | 0.129   | 0.174 |
> | XGBoost | Llama-4-Scout | 0.549 | 0.153 | 0.210 | -0.072 | 0.411 | 0.370 | 0.556 | 0.180  | 0.294 | 0.112   | 0.161 |
> | MLP     | Gemma-3-27B   | 0.467 | 0.332 | 0.116 | 0.085  | 0.507 | 0.486 | 0.599 | -0.053 | 0.296 | -10.632 | 0.218 |
> | MLP     | Llama-4-Scout | 0.413 | 0.247 | 0.192 | 0.005  | 0.317 | 0.402 | 0.610 | -0.155 | 0.309 | -1.042  | 0.102 |

---

> > ### Comment · Reviewer_gfN8 · 2025-11-26
> >
> > Thank you for the detailed rebuttal and additional results. In short, most of my concerns have been addressed, and I will keep my positive score.

---

> > > ### Author Response · Authors · 2025-11-27
> > > **Response to Reviewer gfN8**
> > >
> > > Thank you for the kind follow-up and for recognizing our revisions. We're glad that most of your concerns have been addressed. If you feel the additional experiments and clarifications have strengthened the paper further, we would sincerely appreciate your consideration in adjusting the score accordingly. Please don't hesitate to let us know if you have any remaining questions or suggestions.

---

### Official Review · Reviewer_pPnJ · 2025-10-31

**Soundness:** 3
**Presentation:** 3
**Contribution:** 3
**Rating:** 6
**Confidence:** 4

**Summary:**

The paper presents CityLens, a comprehensive benchmark for evaluating Large Vision-Language Models (LVLMs) on urban socioeconomic sensing using satellite and street-view imagery. Covering 17 cities, 11 indicators, and six domains, CityLens defines three evaluation paradigms and benchmarks 17 LVLMs. Results show that while LVLMs capture some visually grounded socioeconomic patterns, they perform poorly on abstract or weakly visual indicators, highlighting key limitations and opportunities for future research in multimodal urban analytics.

**Strengths:**

1. This paper proposes CityLens, a large-scale and multi-domain benchmark designed to evaluate the performance of LVLMs on urban socioeconomic indicator prediction tasks. It represents a pioneering attempt in this research area.
2. The dataset covers 17 cities, 11 socioeconomic indicators, and 17 LVLMs, with a large experimental scale and carefully designed data collection, mapping, and preprocessing pipelines. The paper introduces three distinct evaluation paradigms—Direct Metric Prediction, Normalized Estimation, and Feature-Based Regression—providing a multi-perspective analytical framework. Both data and code are publicly available, ensuring strong reproducibility and potential academic impact.
3. Beyond comparing model performance, the paper also investigates the effects of geographic variation, input modality (satellite vs. street view), the number of visual features, and Chain-of-Thought (CoT) reasoning strategies.

**Weaknesses:**

1. The results in the Direct Metric Prediction and Normalized Estimation sections (with most tasks showing R² below 0.2 or even negative) clearly demonstrate the severe limitations of current LVLMs in numerical prediction tasks. From a methodological perspective, relying on prompting to have LVLMs directly output numerical values may be inherently unstable. Such a mechanism is fundamentally ill-suited for precise regression tasks.
2. The current experiments primarily focus on comparing multiple general-purpose LVLMs (e.g., GPT-4o, Gemini, Qwen2.5VL), which are not specifically optimized for urban or geospatial applications. As a result, the findings mainly reflect the deficiencies of generic LVLMs on urban tasks rather than revealing the key factors that could improve model performance.
3. Although the paper presents extensive experimental results, it lacks deeper mechanistic analysis explaining why models perform poorly and which visual or linguistic factors contribute to these errors. For instance, indicators with strong visual correlations such as Building Height and GDP perform relatively well, whereas tasks like Life Expectancy and Mental Health perform extremely poorly—an observation that deserves more in-depth discussion.

**Questions:**

Has the author considered including domain-specific models such as UrbanVLP, UrbanCLIP, or UrbanGPT, as well as other fine-tuned LVLMs, as part of the baselines? These models incorporate spatial alignment or social semantic injection mechanisms during training, which may better align with the research objectives of this paper. Analyzing how their training paradigms affect the results would provide deeper insights and elevate the work from a mere model evaluation study to a training paradigm analysis at the methodological level.

---

> ### Author Response · Authors · 2025-11-21
> **Author Response to Reviewer pPnJ (part 1/2)**
>
> We are delighted that the reviewer considers our work to represent a pioneering attempt in this research area. We also thank the reviewer for all the constructive feedback, which helped us revise our paper to make it stronger. In the revised version of the manuscript, we have updated the paper accordingly and highlighted all changes in blue for your convenience. We hope the following additional experiments and clarifications can address the reviewer's remaining concerns:
>
> > **For W1 & W2 & Q1: Evaluation of domain-specific models**
>
> Our benchmark results confirm that predicting socioeconomic indicators as precise regression targets remains highly challenging for current general-purpose LVLMs. We appreciate the reviewer's insightful suggestion regarding the use of domain-specific models, and in response, we conduct new baseline experiments incorporating three recently proposed urban-specific models: UrbanCLIP [1], UrbanVLP [2], and UrbanLLaVA [3]. We would like to clarify that UrbanGPT [4] primarily targets spatio-temporal sequence forecasting tasks, which differ significantly from our current formulation focused on visual-based socioeconomic sensing. Therefore, we did not include it as a direct baseline. For UrbanCLIP and UrbanVLP, we follow the Feature-Based Regression setup, using the same training–testing split. UrbanLLaVA is the most recent domain-specific LVLM, trained with supervised fine-tuning (SFT) on multimodal urban data. We fine-tune Qwen2.5-VL-7B, which is the same base model used in UrbanLLaVA, on additional CityLens data not included in the benchmark. It is important to note that UrbanLLaVA's training data does not include samples for three indicators: House Price, Violent Crime, and Life Expectancy. Therefore, we do not evaluate this model on these three tasks. The results of these additional experiments are summarized in the following table. We have also updated Tables 1 and 2 in the latest version of the paper to include these new results.
>
> The performance of the three domain-specific models reveals notable differences that reflect their underlying training strategies. UrbanCLIP, which relies on contrastive vision–language pretraining using only satellite imagery, shows relatively weak performance across most indicators, such as 0.030 on the Population task and 0.021 on Mental Health. UrbanVLP incorporates both street view and satellite imagery, along with spatial positional encodings in a multi-granularity contrastive learning framework. It outperforms UrbanCLIP across almost all indicators, highlighting the importance of integrating multi-modal and multi-scale urban visual signals for more robust representation learning. UrbanLLaVA, which is trained via supervised fine-tuning on multimodal urban data, achieves the best overall performance, especially on complex tasks like Mental Health and Bachelor Ratio. Its end-to-end LVLM architecture allows for reasoning directly over diverse visual and textual input without relying on handcrafted feature extraction or separate regressors. These findings highlight the promising potential of LVLMs for urban socioeconomic sensing. They reinforce our central motivation for benchmarking such models in this domain and underscore the importance of developing domain-specific LVLMs to address this real-world and socially significant challenge.
>
>
> | Method                        | GDP   | Pop.  | HP    | VC    | PT    | DR    | BH    | MH    | AH    | LE    | BR    |
> | ----------------------------- | ----- | ----- | ----- | ----- | ----- | ----- | ----- | ----- | ----- | ----- | ----- |
> | Best Feature-Based Regression | 0.517 | 0.366 | 0.243 | 0.110 | 0.567 | 0.525 | 0.590 | 0.229 | 0.310 | 0.263 | 0.311 |
> | UrbanCLIP                     | 0.450 | 0.030 | 0.316 | 0.033 | 0.128 | 0.123 | 0.612 | 0.021 | 0.191 | 0.024 | 0.094 |
> | UrbanVLP                      | 0.717 | 0.132 | 0.559 | 0.149 | 0.551 | 0.446 | 0.807 | 0.403 | 0.382 | 0.025 | 0.422 |
> | UrbanLLLaVA                   | 0.679 | 0.320 | /     | /     | 0.583 | 0.626 | 0.887 | 0.449 | 0.387 | /     | 0.556 |
>
> [1] Yan, Yibo, et al. "UrbanCLIP: Learning Text-enhanced Urban Region Profiling with Contrastive Language-Image Pretraining from the Web." WWW 2024
>
> [2] Hao, Xixuan, et al. "UrbanVLP: Multi-Granularity Vision-Language Pretraining for Urban Socioeconomic Indicator Prediction." AAAI 2025
>
> [3] Feng, Jie, et al. "UrbanLLaVA: A Multi-modal Large Language Model for Urban Intelligence." ICCV 2025
>
> [4] Li, Zhonghang, et al. "UrbanGPT: Spatio-Temporal Large Language Models." KDD 2024

---

> ### Author Response · Authors · 2025-11-21
> **Author Response to Reviewer pPnJ (part 2/2)**
>
> > **For W3: Error Cases of LVLMs on Challenging tasks**
>
> We thank the reviewer for this insightful suggestion, which helped improve the completeness of our paper. In response, and also incorporating related feedback from Reviewer gfN8, we have added a new paragraph in "**Section 3.5: Error Analysis and Upper Bounds of LVLMs**" to qualitatively analyze failure cases on challenging tasks such as Mental Health. The corresponding text is also included below. (Due to format constraints, figures are not shown here but are included in the paper.)
>
> `Error Cases on Challenging tasks` To better understand why LVLMs underperform on challenging tasks, we analyze representative errors observed under both the Feature-Based Regression and CoT Prompting setups. As illustrated in Figure 8, errors can arise from both visual perception and linguistic reasoning. For example, during feature extraction, Gemma-3-4B fails to detect small but meaningful elements such as street signs, hallucinates non-existent persons, and underestimates visible greenery, assigning a low score to "Grass \& Shrubs". These errors reveal a lack of fine-grained visual grounding and semantic alignment, which can propagate into downstream reasoning and prediction. We also observe reasoning errors in the CoT setting. For instance, in a Mental Health prediction case, the model overly focuses on a few old and modest houses while overlooking numerous well-maintained, even upscale beachfront apartments visible in the street view images. Moreover, it fails to leverage the region’s proximity to water, which is a known factor with strong aesthetic and calming effects that are directly linked to mental well-being. This suggests that current LVLMs may struggle to appropriately weigh holistic environmental cues during reasoning.

---

### Official Review · Reviewer_DkC7 · 2025-11-01

**Soundness:** 4
**Presentation:** 3
**Contribution:** 3
**Rating:** 6
**Confidence:** 4

**Summary:**

This paper presents CityLens—a comprehensive benchmark designed to evaluate the ability of Large Vision-Language Models (LVLMs) to predict socioeconomic indicators from satellite and street-view imagery. The benchmark covers 17 cities worldwide, spans six key domains—economy, education, crime, transportation, health, and environment—and defines 11 prediction tasks, assessed under three evaluation paradigms: Direct Metric Prediction, Normalized Metric Estimation, and Feature-Based Regression. The study systematically evaluates 17 state-of-the-art LVLMs, making CityLens the most extensive benchmark to date in terms of geographic coverage, indicator diversity, and model scale. Results show that while LVLMs demonstrate promising perceptual and reasoning capabilities, they still face significant limitations in accurately predicting real-world socioeconomic indicators. CityLens provides a unified framework for diagnosing these challenges and advancing future research in urban intelligence and multimodal socioeconomic analysis.

**Strengths:**

1. CityLens introduces the most extensive benchmark to date for evaluating LVLMs on urban socioeconomic prediction, with unprecedented coverage across 17 globally distributed cities, 11 diverse tasks, and 6 critical socioeconomic domains. This scale and diversity significantly advance beyond prior urban vision or geospatial AI benchmarks.

2. Beyond reporting performance numbers, the paper offers thoughtful discussion on why LVLMs struggle—such as lack of numerical grounding, sensitivity to visual ambiguity, and collapse to city-level averages—thereby guiding future model design and training strategies for urban perception tasks.

**Weaknesses:**

Please refer to questions.

**Questions:**

1. I am quite puzzled about the basic motivation of this paper: why choose to predict socioeconomic indicators by feeding street-view or satellite images into large vision–language models (LVLMs)? The existing literature already offers numerous methods specifically designed for modeling urban regions (e.g., region representation learning). I suspect such specialized models may substantially outperform general-purpose multimodal large models in predictive accuracy. Could the authors further clarify the theoretical or practical rationale for choosing LVLMs over traditional urban computing models?

2. I find it difficult to understand how an LVLM can effectively infer complex socioeconomic indicators solely from street-view images. LVLMs were originally designed for semantic understanding of image content (e.g., object recognition, scene description), and their training objectives are not explicitly tied to socioeconomic variables. Thus, establishing a reliable mapping from visual features to abstract socioeconomic indicators seems to lack sufficient mechanistic support. Could the authors explain by what internal mechanisms (e.g., implicit knowledge, vision–language alignment, etc.) the LVLM accomplishes this cross-modal reasoning?

3. Street-view imagery may suffer from serious representational limitations. For example, two areas with vastly different levels of economic development may exhibit highly similar architectural styles or street layouts, causing the model to output similar predictions despite substantial differences in the ground-truth indicators; conversely, areas with similar socioeconomic conditions may yield inconsistent predictions if the street-view sampling locations differ (e.g., arterial roads vs. back alleys). Would the noise and bias introduced by the randomness of image sampling severely undermine predictive reliability? Is reliance on street-view images alone sufficient to support robust socioeconomic sensing?

4. In the “Region” column of Figure 3b, the label “sat” appears. Does this denote satellite imagery? If so, how do the authors extract socioeconomic information from region-level satellite images? Compared with street views, what complementary or more critical visual cues do satellite images provide?

5. I am very interested in the results of Figures 5b and 5c. According to recent work such as FlexiReg: Flexible Urban Region Representation Learning, satellite imagery is often more discriminative than street-view imagery for urban representation. Do the results in this paper support that conclusion as well? Could you further explain why satellite imagery might be more effective than street-view imagery for predicting socioeconomic indicators? What differences in spatial scale or semantic information underlie this discrepancy?

6. I consider the “Prompt Design and Case Analysis” section to be one of the most insightful parts of the paper, as it directly reveals how the LVLM is guided to perform structured scoring. Given the methodological importance of these details, I recommend moving this content into the main text rather than the appendix, so that readers can better understand the coupling between model behavior and task design.

7. I am intrigued by the ablation studies on vision foundation models such as DINO and SigLIP presented at the end of the paper. Could the authors elaborate on the functional differences between these pure vision encoders and end-to-end multimodal large models (e.g., LLaVA, Qwen-VL) for this task? Which class of models contributes more to the final predictive performance? Is it reasonable to conclude that high-quality visual representations are the key to success, while the language model primarily serves as a “scoring interface”?

---

> ### Author Response · Authors · 2025-11-21
> **Author Response to Reviewer DkC7 (part 1/4)**
>
> We are delighted that the reviewer finds our experimental results interesting. We also thank the reviewer for all the meaningful and insightful questions. In the revised version of the manuscript, we have updated the paper accordingly and highlighted all changes in blue for your convenience. We hope the following additional experiments and clarifications can address the reviewer's remaining concerns:
>
> > **For Q1: Why use LVLMs for urban socioeconomic sensing**
>
> We thank the reviewer for raising this important and meaningful question. We agree with the reviewer that specialized representation learning models may outperform **general LVLMs** in predictive accuracy on cities and indicators **seen during training**. However, we would like to highlight that LVLMs, particularly those fine-tuned for urban prediction tasks, offer several unique advantages that surpass traditional representation learning approaches:
>
> 1. **Modality Integration.** LVLMs can natively perform modality integration, seamlessly processing and fusing diverse visual data sources such as street view images and satellite imagery [1, 2]. In contrast, traditional representation learning methods typically require handcrafted feature extraction pipelines or contrastive learning strategies to align multi-modal data.
> 2. **Commonsense and World Knowledge.** LVLMs, by virtue of their language-centric architecture and large-scale pretraining, bring built-in commonsense reasoning and a broad worldview. These models naturally accumulate knowledge related to socioeconomic indicators [3, 4], which implicitly supports their predictive capabilities for urban environments.
> 3. **Universal Modeling Capability.** Unlike traditional representation learning approaches, which usually require a separate model for each city or each socioeconomic indicator [5,6], LVLMs serve as universal models capable of predicting multiple indicators across diverse cities within a single unified framework [7, 8].
> 4. **Region and Indicator Generalization.** Recent studies on fine-tuned LVLMs for urban prediction tasks (e.g., [9], [10]) have demonstrated that LVLMs exhibit strong generalization to unseen regions and indicators. This is likely due to their accumulated global knowledge and transferable prediction paradigm, which are capabilities that traditional models typically lack, and as a result, they often perform well only on cities or indicators seen during training.
> 5. **Explainable Predictions.** Another unique advantage of LVLMs is their capacity for explainable prediction. LVLMs can articulate the rationale behind their outputs, providing human-readable justifications or reasoning chains for each prediction [8, 9, 10], which is an important property not available in most traditional representation learning methods.
>
> We hope this clarification helps convey our motivation for leveraging LVLMs in this context, and why we believe they offer distinct and valuable benefits over conventional models for urban socioeconomic sensing.
>
> [1] Chen, Joya, et al. "VideoLLM-online: Online Video Large Language Model for Streaming Video." CVPR 2024
>
> [2] Liu, Haotian, et al. "Visual Instruction Tuning." NeurIPS 2023
>
> [3] Nitzan Bitton-Guetta, et al. "Visual Riddles: a Commonsense and World Knowledge Challenge for Large Vision and Language Models." NeurIPS 2024
>
> [4] Hou, Ce, et al. "Urban sensing in the era of large language models." Innovation 2025
>
> [5] Yan, Yibo, et al. "UrbanCLIP: Learning Text-enhanced Urban Region Profiling with Contrastive Language-Image Pretraining from the Web." WWW 2024
>
> [6] Hao, Xixuan, et al. "UrbanVLP: Multi-Granularity Vision-Language Pretraining for Urban Socioeconomic Indicator Prediction." AAAI 2025
>
> [7] Zhang, Xin, et al. "UrbanMLLM: Joint Learning of Cross-view Imagery for Urban Understanding." (2025).
>
> [8] Zhang, Yunke, et al. "Perceiving Urban Inequality from Imagery Using Visual Language Models with Chain-of-Thought Reasoning." WWW 2025
>
> [9] Liu, Tianhui, et al. "CityRiSE: Reasoning Urban Socio-Economic Status in Vision-Language Models via Reinforcement Learning." arXiv preprint arXiv:2510.22282 (2025).
>
> [10] Wang, Qiongyan, et al. "Urban-R1: Reinforced MLLMs Mitigate Geospatial Biases for Urban General Intelligence." arXiv preprint arXiv:2510.16555 (2025).

---

> ### Author Response · Authors · 2025-11-21
> **Author Response to Reviewer DkC7 (part 2/4)**
>
> > **For Q2: Hypothesized mechanisms behind LVLMs' cross-modal reasoning from visual inputs to socioeconomic indicators**
>
> We thank the reviewer for raising this important and valuable question, which we have also carefully considered during the experimental design and analysis stages of our study.
>
> To clarify, as described in Section 2.1 and Appendix A.6.1, under the setting of 'Direct Metric Prediction' and 'Normalized Metric Estimation', where the LVLM is used directly as a predictor to estimate socioeconomic indicators, the model's input is not limited to street view images alone. Instead, it incorporates **street view images, satellite imagery, and the name of the city** in which the region is located. This combination provides the LVLM with diverse types of spatial and contextual information, ground-level, top-down, and semantic, which together enhance the model's capacity for reasoning about socioeconomic conditions.
>
> For the reviewer's concern about the underlying mechanisms of cross-modal reasoning in LVLMs, as discussed in **Appendix A.10: **Hypotheses on LVLMs for Urban Socioeconomic Sensing****, we propose a hypothesis about the internal mechanism by which LVLMs might link visual inputs to socioeconomic indicators:
>
> We propose to view visual concepts along two dimensions: low-level and high-level.
>
> * **Low-level concepts** refer to basic, concrete visual features such as greenery, street furniture, or vehicle density, which are often directly observable and easily recognized by traditional machine learning models. These features correspond to what the reviewer described as the LVLMs' capability for semantic understanding of image content, including object recognition and scene description.
> * **High-level concepts**, in contrast, such as signs of poverty, infrastructure quality, or commercial activity are abstract constructs that emerge from the combination of multiple visual and contextual cues. These concepts are more closely entangled with socioeconomic meaning and are typically associated with human-level interpretation and reasoning.
>
> One of the key strengths of LVLMs lies in their ability to go beyond surface-level visual recognition, and to **implicitly perceive and reason about these high-level visual semantics**.
>
> When presented with a socioeconomic prediction task, an LVLM may follow a multi-level reasoning process. It first identifies high-level visual concepts that are semantically associated with the target indicator. For example, to estimate the bachelor ratio, the model may consider abstract cues such as signs of poverty, commercial activity, and various other concepts. To support the recognition of these high-level visual concepts, the model must detect corresponding low-level visual features from the input images such as building density, greenery coverage, and so on. These low-level features are more concrete and directly extractable from satellite or street view images. The model then aggregates these features into high-level visual concepts, which are further cross-checked and composed through reasoning. Before producing a final prediction, the LVLM may also draw upon its world knowledge, learned during pretraining. For instance, the model may recall the typical range or distribution of bachelor ratios for a given city or country, and use this prior in combination with the inferred visual semantics to produce a more calibrated final answer.

---

> ### Author Response · Authors · 2025-11-21
> **Author Response to Reviewer DkC7 (part 3/4)**
>
> > **For Q3: Street view imagery representational limitations**
>
> Under the setting of 'Direct Metric Prediction' and 'Normalized Metric Estimation', our model input for each region includes ten street view images, one satellite image, and the name of the city in which the region is located. The task is to predict a specific socioeconomic indicator for that region. We acknowledge the reviewer's concern that two areas with vastly different levels of economic development may, in some cases, exhibit highly similar architectural styles or street layouts. Indeed, in our own experiments, we observed that street views in central Cape Town are visually very similar to those found in many Western cities.
>
> However, our input design goes beyond street view images: the inclusion of **satellite imagery**, which provide a broader spatial context, and the **city name**, which allows the LVLM to activate relevant **world knowledge** about the socioeconomic characteristics of that city, significantly reduces the ambiguity mentioned by the reviewer. With these additional sources of information, we believe the concern raised is likely to affect only a small fraction of cases in practice.
>
> Moreover, several prior studies like [1] have shown that **street view images alone** can already serve as a strong predictor for socioeconomic indicators, often outperforming other data sources such as POI-based features. This further validates the effectiveness of visual urban features in socioeconomic sensing, even in the absence of extensive contextual input.
>
> [1] Fan, Zhuangyuan, et al. "Urban visual intelligence: Uncovering hidden city profiles with street view images." PNAS 2023
>
> > **For Q4: Visual cues provided by satellite imagery**
>
> As described in Section 2.1 Dataset Construction, the "sat" label in the "Region" column of Figure 3b denotes a "satellite area", meaning that the "region" corresponding to the prediction task is defined by the spatial extent of a satellite image. More specifically, as detailed in Appendix A.5.2 Data Mapping and Aggregating, for tasks where "sat" defines the region type, the raw data typically comes from GeoTIFF files. In these cases, we use **the area covered by a single satellite image as the region unit**. The ground-truth value of the target socioeconomic indicator is computed by aggregating the raw data over that satellite-covered area. And the visual input to the model includes the satellite image and 10 street view images sampled from within that satellite-covered area.
>
> Street view images capture fine-grained, semantically rich visual details that closely align with human perception. These details are often highly informative for socioeconomic prediction. However, street view data is typically **sparse and localized**, covering only a limited number of points within a region. As such, it may suffer from sampling bias or fail to represent the overall characteristics of the area.
>
> In contrast, satellite imagery provides a **macroscopic spatial overview** of the entire region, capturing features such as:
>
> * Building density and spatial distribution
> * Road network structure and regularity
> * Proximity to the city center or periphery
> * Dominant building types (low-rise or high-rise)
> * Green coverage or open spaces
>
> These aggregated spatial patterns are difficult to infer from sparse street level images, yet they are often strongly associated with socioeconomic conditions. Satellite imagery thus enables the model to form an **overall impression** of the region and provides **stable, unbiased context** that complements the detailed but localized information from street views.
>
> Street view and satellite imagery offer complementary visual perspectives that, when combined, provide a more comprehensive understanding of the urban environment for socioeconomic prediction.

---

> ### Author Response · Authors · 2025-11-21
> **Author Response to Reviewer DkC7 (part 4/4)**
>
> > **For Q5: Respective roles of street view and satellite imagery**
>
> We extend the analysis in Figure 6b by conducting additional ablation experiments where we remove street view images and use only satellite imagery as input. The results are shown in the table below. We have also updated the corresponding section in the paper to include these new results, with the changes highlighted in blue for easy reference. We have included the revised text below for your convenience.
>
> `Impact of Input Modalities.` In this part, we evaluate the impact of input modalities by comparing model performance in three configurations: using both, only street view, and only satellite imagery. We test House price, Public transport, and Drive ratio using Gemini-2.0-Flash under the Direct Metric Prediction setting. Contrary to prior findings that satellite imagery is often more discriminative than street view imagery for urban representation [1, 2], our results in Figure 6b show that using street view images alone achieves performance comparable to using both street view and satellite imagery, and significantly outperforms using satellite imagery alone. This suggests that street view images provide more semantically rich and fine-grained visual cues, such as building façades, commercial signage, and infrastructure quality. These ground-level features are likely more tightly coupled with socioeconomic indicators and more readily interpreted by current LVLMs, which have been pretrained extensively on image–language pairs featuring such localized, human-centric content. While satellite imagery exhibits weaker predictive performance, it still contributes independent spatial context, such as urban morphology and building layout. However, it may not offer the same level of semantic density at the resolution used in CityLens.
>
> Additionally, we would like to further clarify during the rebuttal stage that in our setting where LVLMs are used to predict socioeconomic indicators from visual input, street view imagery contributes more significantly than satellite imagery, which contrasts with conclusions from prior work such as FlexiReg [1]. We believe this discrepancy is reasonable and stems from fundamental differences in **task objectives and model architectures**. FlexiReg focuses on urban representation learning, where satellite imagery may indeed be more discriminative due to its ability to capture large-scale spatial layouts and structural patterns. In contrast, our task may require interpretable, localized visual cues that are more effectively conveyed through street view imagery and more readily leveraged by LVLMs.
>
>
> | Task    | Input Modality          | $R^2$ |
> | ----------- | ----------------------- | ----- |
> | House Price | Street view + Satellite | 0.373 |
> | | Street view     | 0.374 |
> || Satellite    | 0.249 |
> | Drive       | Street view + Satellite | 0.561 |
> | | Street view | 0.567 |
> | | Satellite| 0.160 |
> | Public      | Street view + Satellite | 0.395 |
> || Street view  | 0.376 |
> || Satellite  | 0.277 |
>
> [1] Sun, Fengze, et al. "FlexiReg: Flexible Urban Region Representation Learning." KDD 2025
>
> [2] Hao, Xixuan, et al. "UrbanVLP: Multi-Granularity Vision-Language Pretraining for Urban Socioeconomic Indicator Prediction." AAAI 2025
>
> > **For Q6: Moving "Prompt Design and Case Analysis" to the main text**
>
> We thank the reviewer for this valuable and constructive suggestion, which has helped make our paper more robust. In the revised version, we have added Figure 4, which presents prompt examples for the three evaluation methodologies, to help readers better understand the design and usage of each setting.
>
> > **For Q7: Roles of Vision Encoders and Language Models**
>
> An end-to-end LVLM is typically composed of two major components: a vision encoder and a large language model (LLM). In the context of urban socioeconomic sensing, we believe that **both components play distinct and important roles**.
>
> The vision encoder is responsible for extracting visual representations from the input images. This forms the foundation of the model's perception of the urban environment, capturing information from both street view and satellite imagery. The LLM, on the other hand, contributes in ways that go beyond visual encoding. It brings in commonsense reasoning abilities, generalized world knowledge, and task-general inference capabilities. These are especially valuable in our task, where making predictions about socioeconomic indicators often requires linking visual patterns to abstract concepts, grounding them in real-world context.
>
> In this sense, while high-quality visual representations are crucial, the LLM is not merely a scoring interface. Rather, it actively participates in interpreting, reasoning over, and contextualizing the visual input.

---

### Author Response · Authors · 2025-11-30
**Response Summary (part 2/2)**

Finally, we kindly ask the Area Chair to consider Reviewer Zxk9's review with caution, as we believe the reviewer's comments **exhibit noticeable bias**. This concern is further reflected in the score of 2, which contrasts sharply with the positive scores of 6 given by all other reviewers.

* W1: The reviewer comments that "the paper's central contribution is a new benchmark, which is not in line with the scope of ICLR." However, this statement is inconsistent with ICLR's established scope. ICLR explicitly lists "datasets and benchmarks" as a primary subject area, and in past years, multiple dataset and benchmark papers have been accepted. Moreover, beyond what the reviewer describes as proposing a new benchmark, we also release a curated dataset, where we collect, process, and match socio-economic indicators with urban imagery from multiple original sources. The data used in the benchmark is a subset of this dataset, so we believe our contribution aligns well with ICLR's 'datasets and benchmarks' area.
* W3: The reviewer asserts that our findings on CoT and vision encoders are "well-established consensus." However, we do not share this view, and in the rebuttal process, we provide several examples to address and counter this claim. Notably, Reviewer gfN8 asks us to explain why CLIP performs best among the evaluated vision encoders, which further indicates that our findings are not self-evident, contrary to Reviewer Zxk9's assertion.
* W4: The reviewer asserts that our "evaluation protocols comparison is unfair". We believe the reviewer has misunderstood our experimental setup, and we have further clarified the details of our task configuration. We do not consider this point to be a valid weakness of our work.
* W5 & Q1: The reviewer states that "the conclusion that 10 ground-level images dominate the information from one overhead image is unsurprising". We respectfully disagree with this point, as we believe the reviewer has misunderstood the information that satellite images and street view images each provide. We provide support from a relevant previous paper that contradicts the reviewer's claim. Moreover, Reviewer DkC7 references the FlexiReg [1] paper, which concludes that satellite imagery is often more discriminative than street view imagery for urban representation, and asks whether the same holds in the LVLM-based socioeconomic sensing setting. This clearly suggests that Reviewer DkC7 does not consider our conclusion to be self-evident or unsurprising, contrary to Reviewer Zxk9's assertion.

Overall, the reviewer Zxk9 raises five weaknesses and one question, with the question being closely related to weaknesses 5. We strongly disagree with four of the five points (W1, W3, W4, W5), which we believe stem from misunderstandings or subjective bias. This is further supported by the fact that the other reviewers provide feedback that directly contradicts reviewer Zxk9's statement. For the one remaining point (W2), we appreciate the suggestion and have updated the paper accordingly by introducing Figure 8, which showcases examples of visual cues that models misinterpret or fail to utilize, and providing a more detailed qualitative analysis of why models struggle with challenging tasks like Mental Health. Regarding Q1, we have conducted additional experiments using only satellite imagery as visual input, and we discuss this in conjunction with W5 to further elaborate on our findings.

We would like to once again express our sincere gratitude to the Area Chair for carefully reviewing our paper and rebuttal. We truly appreciate Area Chair's time and thoughtful consideration.

Best regards,

Authors of CityLens

[1] Sun, Fengze, et al. "FlexiReg: Flexible Urban Region Representation Learning." KDD 2025

---

### Author Response · Authors · 2025-11-30
**Response Summary (part 1/2)**

Dear Area Chair and Reviewers,

We sincerely thank Area Chair for the time and consideration of our paper *CityLens*. We greatly appreciate the constructive feedback and suggestions from all the reviewers. We are delighted that Reviewer DkC7 finds our experimental results interesting, Reviewer pPnJ considers our work to represent a pioneering attempt in this research area, and Reviewer gfN8 finds our paper well-organized, the experimental results comprehensive, and the contributions valuable to the community. Initially, our paper received scores of **6, 6, 6, and 2**. During the brief rebuttal period, only Reviewer gfN8 responded to us, stating that most concerns had been addressed and that the reviewer would **maintain original positive score of 6**. We are delighted that our responses resolved reviewer's concerns.

In line with the initial feedback from the reviewers, we have made updates to the manuscript, with the changes highlighted in blue. Here we summarize the revision as follows:

1. We extend the analysis in Figure 6b by conducting additional ablation experiments using only satellite imagery as visual input. We also update Section 3.3 to include a discussion on the 'Impact of Input Modalities', as suggested by Reviewer DkC7 and Zxk9.
2. We add Figure 4 in Section 2.2, which presents prompt examples for the three evaluation methodologies, to help readers better understand the design and usage of each setting, as suggested by Reviewer DkC7.
3. We add the performance of domain-specific models, including UrbanCLIP, UrbanVLP, and fine-tuned LVLMs, on the CityLens benchmark. The results are now included in Table 1 and Table 2 of the paper, as suggested by Reviewer pPnJ and gfN8.
4. We introduce Figure 8 to showcase examples of visual cues that models misinterpret or fail to utilize, and provide a more detailed qualitative analysis of why models struggle with challenging tasks like Mental Health, as suggested by Reviewers pPnJ, gfN8, and Zxk9.
5. We add the results of nonlinear regressors, including Random Forest, XGBoost, and MLP, to regress the 13-attribute vectors, and place the details in Appendix A.4 due to page limitations, as suggested by Reviewer gfN8.

In addition to the experimental changes in the paper, we have also provided detailed responses to the explanatory questions raised by the reviewers:

* For Reviewer DkC7: We provide a detailed explanation of why we use LVLMs for urban socioeconomic sensing, emphasizing their advantages over traditional representation learning methods. We also elaborate on the hypothesized mechanisms behind LVLMs' cross-modal reasoning from visual inputs to socioeconomic indicators. Additionally, we clarify the setup of our task and the distinct types of visual information that satellite and street view imagery can each provide. Finally, we share our insights on the roles of Vision Encoders and Language Models in urban socioeconomic sensing tasks, explaining how each component contributes to the overall process.
* For Reviewer gfN8: We provide our explanation for CLIP's superior performance among the four vision encoders we evaluate.

---

### Meta-Review · Area_Chair_FRXL · 2026-01-13

**Summary:**

Reviewers agreed that the benchmark is valuable, timely, and well executed in terms of scale, coverage, and reproducibility. The main concerns were about clarifying the motivation for using LVLMs, the stability of direct numeric prediction via prompting, the need for domain-specific or fine-tuned baselines, deeper qualitative error analysis, and clearer presentation of modality comparisons and evaluation protocols. These concerns were addressed in the rebuttal and revision, and the remaining differences mainly reflect perspective on contribution type rather than technical shortcomings.

**Reviewer Concerns:**

The rebuttal addressed most concerns about motivation, clarity, and empirical completeness, including the rationale for using LVLMs, the hypothesized cross-modal reasoning mechanism, the roles of street-view versus satellite imagery, the inclusion of domain-specific and fine-tuned baselines, the addition of modality ablations and nonlinear regressors, and the expansion of qualitative error analysis with concrete failure cases. These points largely resolve the concerns raised by Reviewers DkC7, pPnJ, and gfN8.

The main outstanding concern is Reviewer Zxk9’s objection regarding the scope and nature of the contribution (benchmark vs. method) and their framing of the evaluation as unfair or unsurprising. However, the authors have clarified the scope, positioning, and intent of the work, and the remaining disagreement appears to be largely about contribution framing rather than unresolved technical issues. Overall, the rebuttal sufficiently addressed the substantive concerns raised during review.

**Reviewer Scores:**

Overall, the discussion would likely have led the three positive or borderline reviewers to maintain or slightly increase their scores, while the dissenting reviewer Zxk9 might increase slightly but would likely remain the most critical.

---

### Decision · Program_Chairs · 2026-01-26

Accept (Poster)